# Seasonal soil moisture and crop yield prediction with SEAS5 long-range meteorological forecasts in a land surface modelling approach

Theresa Boas[1,2,3], Heye Reemt Bogena[1], Dongryeol Ryu[3], Harry Vereecken[1,2], Andrew Western[3], Harrie-Jan Hendricks Franssen[1,2]

[1]Research Centre Jülich, Institute of Bio- and Geosciences: Agrosphere (IBG-3), 52425 Jülich, Germany
[2]Centre for High-Performance Scientific Computing in Terrestrial Systems: HPSC TerrSys, Geoverbund ABC/J, 52425 Jülich, Germany.
[3]University of Melbourne: Department of Infrastructure Engineering, Parkville VIC 3010, Australia

*Correspondence to:* Theresa Boas (t.boas@fz-juelich.de)

**Abstract.** Long-range weather forecasts provide predictions of atmospheric, ocean and land surface conditions that can potentially be used in land surface and hydrological models to predict the water and energy status of the land surface or in crop growth models to predict yield for water resources or agricultural planning. However, the coarse spatial and temporal resolutions of available forecast products have hindered their widespread use in such modelling applications that usually require high resolution input data. In this study, we applied sub-seasonal (up to 4 months) and seasonal (7 months) weather forecasts from the latest European Centre for Medium-Range Weather Forecasts (ECMWF) seasonal forecasting system (SEAS5) in a land surface modelling approach using the Community Land Model version 5.0 (CLM5). Simulations were conducted for 2017-2020 forced with sub-seasonal and seasonal weather forecasts over two different domains with contrasting climate and cropping conditions: the German state of North Rhine-Westphalia (DE-NRW) and the Australian state of Victoria (AUS-VIC). We found that after preprocessing of the forecast products (i.e. temporal downscaling of precipitation and incoming shortwave radiation), the simulations forced with seasonal and sub-seasonal forecasts were able to provide a model output that was very close to the reference simulation results forced by reanalysis data (the mean annual crop yield showed a maximum difference of 0.28 and 0.36 t/ha for AUS-VIC and DE-NRW, respectively). Differences between seasonal and sub-seasonal experiments were insignificant. The forecast experiments were able to satisfactorily capture recorded inter-annual variations of crop yield. In addition, they also reproduced the generally higher inter-annual differences in crop yield across the AUS-VIC domain (approximately 50 % inter-annual differences in recorded yields and up to 17 % inter-annual differences in simulated yields) compared to the DE-NRW domain (approximately 15 % inter-annual differences in recorded yields and up to 5 % in simulated yields). The high and low yield seasons (2020 and 2018) among the four simulated years were clearly reproduced in forecast simulation results. Furthermore, sub-seasonal and seasonal simulations reflected the early harvest in the drought year of 2018 in the DE-NRW domain. However, the simulated inter-annual yield variability was lower in all simulations compared to the official statistics. While general soil moisture trends, such as the European drought in 2018, were captured by the seasonal experiments, we found systematic over- and underestimations in both the forecast and the reference simulations compared to the Soil Moisture Active Passive Level-3 soil moisture product (SMAP L3) and the Soil Moisture Climate Change Initiative Combined dataset from the European Space Agency's (ESA CCI). These observed biases of soil moisture as well as the low inter-annual differences of

simulated crop yield indicate the need to improve the representation of these variables in CLM5 to increase the model sensitivity to drought stress and other crop stressors.

## 1    Introduction

Reliable high-resolution seasonal weather forecasting systems can provide important information for a multitude of weather-sensitive sectors, especially for agricultural regions with high inter-annual variability of rainfall patterns that are strongly influenced by El Niño events (Ash et al., 2007; McIntosh et al., 2007; Troccoli, 2010). Information on seasonal rainfall and temperature development can influence agricultural management decisions at the beginning of the growing season and potentially mitigate yield losses related to droughts. However, the relevance and usability of such seasonal forecasts depends on the predicted variables, their accuracy and lead time, and whether they are supplied in user-friendly and content-specific format, for example in combination with other model applications (e.g., crop or land surface models) to assess the expected benefits to the economy or natural resources (Cantelaube and Terres, 2005; Hansen et al., 2006; Ash et al., 2007; McIntosh et al., 2007; Meza et al., 2008). Sub-seasonal (1 to 3 months) and seasonal (up to 7 months lead times) forecasts bridge the gap between short-range weather forecasts and climate predictions and are the most important time periods for model applications and planning purposes, e.g., in agriculture or risk management (Monhart et al., 2018). In the last decade, substantial improvements have been made in numerical weather prediction, especially in short- and medium-range weather forecasts by further model development, data assimilation methods and the incorporation of ensemble prediction in seasonal forecasting systems (Coelho and Costa, 2010; Bauer et al., 2015; Monhart et al., 2018).

In spite of these substantial improvements, there are still considerable challenges in interfacing forecast information from climate to systems science (Coelho and Costa, 2010). For instance, deficiencies remain in the definition and communication of forecast uncertainties (e.g., due to discrepancies between the spatial and temporal resolution of the global weather forecasting system and the regional or local land surface models) and in the lack of available tools, literature and experience for the correct usage and data processing (Coelho and Costa, 2010). Seasonal and sub-seasonal forecasts do not reflect day-to-day weather statistics but rather project general weather trends of the predicted season. This leads to high precipitation biases compared to observations, which is a major limitation for crop models that usually operate on sub-daily time steps in response to precipitation and corresponding soil moisture dynamics. In their study, Monhart et al. (2018) conducted a verification of sub-seasonal forecasts (with 1 month lead time) from the European Centre for Medium-Range Weather Forecasts (ECMWF) against ground based observational time series of 20 years across Europe for precipitation and temperature and performed two different bias correction techniques. They found generally better skill for temperature than precipitation and that the accuracy of both variables improved significantly after station-based bias correction (Monhart et al., 2018). However, McIntosh et al. (2007) evaluated the potential of different forecasting systems for wheat growth in Victoria, Australia, and concluded that even a perfect forecast of the total rainfall amount throughout the growing season is not enough to explain even half of the overall potential of an ideal forecasting system.

The major aim of this study was to evaluate the efficacy and applicability of this state-of-the-art forecasting product for physical and biogeochemical land surface responses and regional crop production in an ecosystem process model approach. To this end, we tested the combination of the Community Land Model version 5 (CLM5)

(Lawrence et al., 2018; 2019) and seasonal forecasts from ECMWFs latest seasonal forecasting system SEAS5 (Johnson et al., 2019). Regional simulations were conducted for two domains with different climate regimes and agricultural characteristics, one covering the state of North Rhine-Westphalia in Germany (DE-NRW), and one the state of Victoria in Australia (AUS-VIC), using sub-seasonal and seasonal forecasts with different lead times as input. In our evaluations we focussed on (1) the model's sensitivity to seasonal changes in weather patterns and their effect on regional vegetation properties, e.g., leaf area index (LAI), evapotranspiration (ET), and crop yield; (2) the representation of the surface soil moisture content; and (3) the overall applicability and potential of seasonal weather forecasts for the prediction of regional agricultural production in model applications such as CLM5. In addition, we addressed the pre-processing steps required for the usage of the SEAS5 product in this model application and briefly discuss the importance of temporal downscaling.

The long-range forecast product generated by the ECMWF SEAS5 system, the fifth generation seasonal forecast system that became operational in November 2017 (Johnson et al., 2019), represents one of the most sophisticated seasonal products available to date. Studies that evaluated the quality of the SEAS5 product globally and for specific regions concluded that it outperforms earlier versions of ECMWF forecast products and can provide useful information for regional agriculture (e.g., Johnson et al., 2019; Wang et al., 2019; Gubler et al., 2020). The prediction performance was found to be highest for maximum temperature over South America (with up to 70% probability that the predictions correctly capture the observed outcomes in the tropics during austral summer) (Gubler et al., 2020) and Australia (Wang et al., 2019). For precipitation the performance was considerably lower and more variable (spatially and temporally) than for temperature (Wang et al., 2019; Gubler et al., 2020). The best forecast performance was observed over regions that are influenced by El Niño where SEAS5 outperformed predictions from statistical relationships at the seasonal scale (Gubler et al., 2020).

The relevance and value of meteorological forecasting systems for agriculture has been evaluated by a number of studies (e.g., Cantelaube and Terres, 2005; Marletto et al., 2007; McIntosh et al., 2007; Semenov and Doblas-Reyes, 2007). In their study, Semenov and Doblas-Reyes (2007) used a stochastic weather generator to obtain site-specific daily weather from seasonal DEMETER predictions. They found that dynamical seasonal forecasts did not improve single site yield predictions with the wheat simulation model compared to approaches based on historical climatology due to their low skill for latitudes higher than 30° for northern and southern hemispheres. Cantelaube and Terres (2005) evaluated an ensemble of seasonal weather forecasts from the DEMETER (European Development of a European Multimodel Ensemble system for seasonal to inTERannual climate prediction) project in a multi-model approach with a crop growth modelling system (CGMS), showing encouraging results for the usage of seasonal forecasts for weather sensitive decision making. Wang et al. (2020) investigated the impact of pre- and early-season El Niño Southern Oscillation (ENSO) related large scale climate signals on wheat yields in Australia. They found that these ENSO signals can have a significant impact on wheat yields in the Australian wheat belt and could explain up to 21% of the yield variation. In another study by Potgieter et al. (2022), the lead time and skill of Australian wheat yield forecasts using seasonal climate forecasts derived from a statistical ENSO-analogue system were compared with using a dynamic general circulation model (GCM). They found that ENSO-derived forecasts showed higher skills at a longer lead time (6 months), with a higher correlation coefficient of 0.48 compared to 0.37 for GCM forecasts, while GCM forecasts provided higher skill at shorter lead times (1-3 months) with a higher correlation coefficient of 0.44 compared to 0.35 for ENSO-analogue forecasts.

Thus, although seasonal weather forecasts have an immense potential for the agricultural sector, i.e. for individual farming decisions, risk management and adaptation strategies for increasing climate variability and extreme weather events in the context of climate change (Calanca et al., 2011), they need to be combined with a measurable

system response via, e.g., crop models or earth system models. Land surface models are our primary tools to simulate water, energy and nutrient fluxes in the terrestrial ecosystem and are broadly applied for different scientific purposes (e.g., Niu et al., 2011; Lawrence et al., 2018, 2019; Lombardozzi et al., 2020; Naz et al., 2019). CLM5 is the latest version of the land component in the Community Earth System Model and offers the possibility of prognostic vegetation state and yield prediction with its new biogeochemistry module (Lawrence et al., 2018;

2019). CLM5 includes a representation of crops and agricultural management (fertilization, irrigation, different crop types) which essential to study the impact of climate change on yield as well as the implications of agriculture for climate change (Lombardozzi et al., 2020). In CLM5, crop productivity is a dynamic nonlinear interaction between meteorological conditions, crop phenology, nutrient dynamics, and water availability in the soil. Thus, a reliable prediction of the soil moisture regime is also essential for the relevance of land surface model applications

for climate change research and is a major source of uncertainty for the simulation of the terrestrial carbon cycle (Trugman et al., 2018).

Another major limitation to the usage of seasonal and sub-seasonal forecasting products for crop or land surface modelling is their coarse spatial and temporal resolution. This problem can be addressed by disaggregating forecast variables using stochastic weather generators (e.g. Hansen et al. 2006), which has already been done for several

crop model approaches (see reviews in Cantelaube and Terres, 2005; Ash et al., 2007; Meza et al., 2008; Ouedraogo et al., 2015).

Despite their potential economic value for agricultural production systems, the quantitative adoption of seasonal climate forecasts from farmers is low, both in Victoria and NRW (e.g., Parton et al., 2019). The Australian Bureau of Meteorology attributed this to insufficient data and evidence about their value and conducted a series of studies

of the potential value of a forecast based on a particular production system and for specific regions and timescales (Hansen, 2002; Hansen et al., 2006). Furthermore, the challenges highlighted above have hindered a widespread application of such long-range forecasts for agriculture, particularly for larger (not site-specific) scales (Coelho and Costa, 2010; Calanca et al., 2011). The lack of user-friendly tools and services that can provide crop-specific information based on seasonal forecasts and account for other economic factors (e.g., political choices, outlook

for crop markets, etc.) represents another constraint.

A thorough review on the economic value of seasonal weather forecasts for agriculture can be found in Meza et al. (2008), Klemm and McPherson (2017), and references therein. For an improved understanding of the value of seasonal forecasts for the agricultural sector, more studies are needed that explore state of the art forecast products and for a larger range of regions (i.e., with high seasonal predictability, large areas of extensive management, rain-

fed). Here, we provide a first feasibility study of the combination of seasonal forecasts from SEAS5 with CLM5, focussing on crop yield and soil moisture predictions on regional scale.

## 2       Material and Methods

### 2.1       Regional domains and surface input data

The CLM5 simulations were carried out in two regional domains, one in West-Europe covering the state of North Rhine-Westphalia in Germany (DE-NRW) and one that covers large parts of the state of Victoria in Australia (AUS-VIC) (Figure 1). The DE-NRW domain is characterized by a very diverse land cover with urban, natural and mixed agricultural areas that are mostly fed by rainwater. The agricultural land cover in DE-NRW is especially abundant in the northern and western part of the domain along with natural vegetation and urban areas. Winter wheat, winter barley, corn, sugar beet and rape seed are the most important cash crops in DE-NRW, mostly rain-fed (Figure 1, BMEL, 2020, 2022). In the southern part of the domain over the Eifel region, forests and grasslands are the dominant land cover. Recently, agricultural yield in this area has been impacted in 2018 and 2019, by a late cold spell (late February – early March 2018) and extreme heat and dry spells in both summers which lead to unusually high spatial variability of yield, especially for cereals (NRW Gov., 2020; BMEL, 2020, 2022). The AUS-VIC domain covers large parts of the Australian wheat belt in the state of Victoria (Figure 1). The land cover is dominated by rain-fed agricultural areas with large paddock sizes of mostly cereal cultivation, with winter wheat being the most important crop, followed by barley and canola (ABARES, 2020; Morse-McNabb et al., 2017), along with large patches of naturally vegetated areas (i.e., pasture, grasslands, native woody cover) and woody horticulture and wood plantations. Unfavourable weather conditions for winter crop farming (i.e. the timing and intensity of early season rainfall events) are clearly reflected in relatively low regional production and yield per area (ABARES, 2020).

For the DE-NRW domain, land cover information was derived from the crop and land cover dataset by Griffiths et al. (2019) that covers Germany at 30-m resolution. This dataset was generated from Sentinel-2A MSI and Landsat-8 OLI observation data from the NASA Harmonized Landsat-Sentinel dataset for the year 2016 (Claverie et al., 2018). Comparison of the derived crop type and land cover map with agricultural reference data showed very good overall accuracy of > 80%, especially for crop types with high abundancies, e.g., cereals, maize and canola (Griffiths et al., 2019). For the AUS-VIC domain the 500 m resolution Moderate Resolution Imaging Spectroradiometer (MODIS) land cover product (Friedl and Sulla-Menashe, 2019) was aggregated to the coarser resolution of 1 km and masked with information from the latest Victorian Land Use Information System (VLUIS) product for the year 2016 (Morse-McNabb et al., 2017). The VLUIS dataset covers the whole state Victoria and contains information on land use and land cover for each cadastral parcel. It is a product of time series analysis of remote sensing data (MOD13Q1 or MYD13Q1 by NASA) and annually collected field data (Morse-McNabb et al., 2017).

For both domains, we used soil texture and soil organic matter information from the global SoilGrids database that provides soil information at seven depths (0, 0.05, 0.15, 0.30, 0.60, 1 and 2 m) at 250 m spatial resolution (Hengl et al., 2017). Other soil parameters, such as the saturated hydraulic conductivity and soil retention parameters were calculated within CLM5 with the pedotransfer function after Cosby et al. (1984). Additional properties of each of the sub-grid land fractions (e.g., properties of urban land cover) were derived from the global CLM5 surface dataset (see Lawrence et al., 2018).

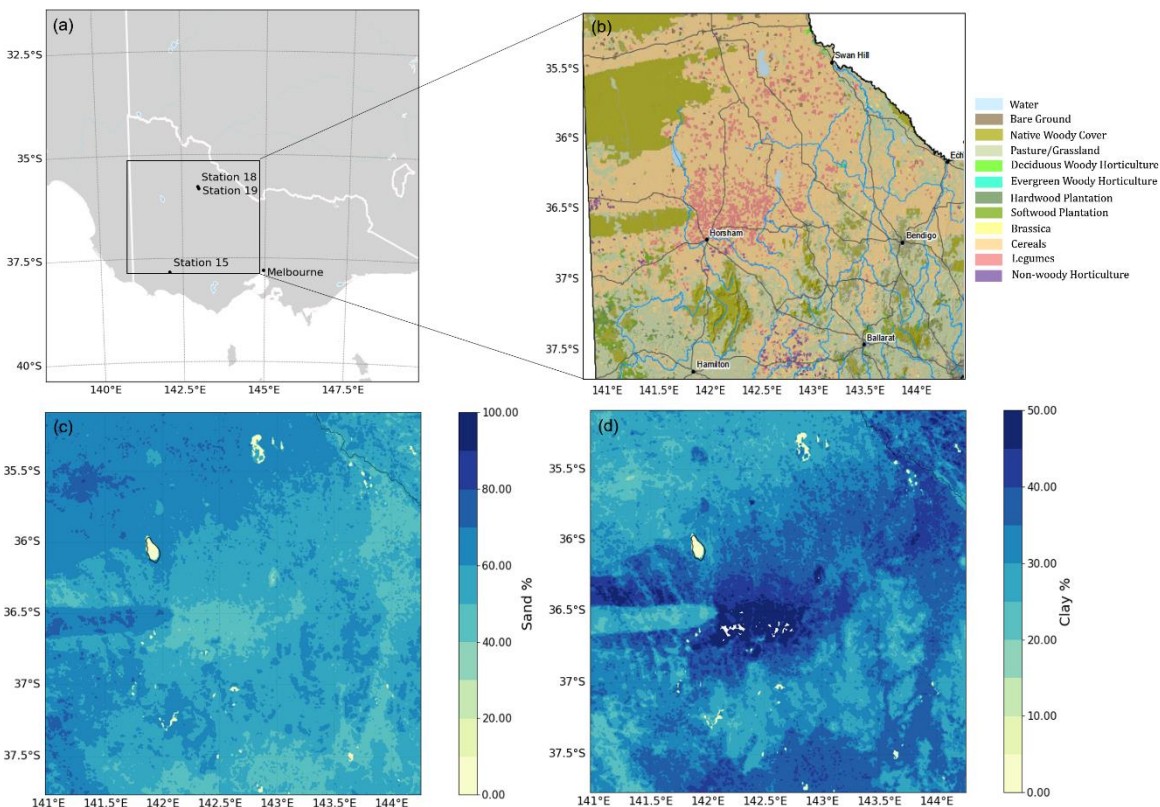

Figure 1: (a) AUS-VIC simulation domain extent, (b) dominant land use type based on VLUIS data, modified after (Victorian Government Data Directory, 2020; Morse-McNabb et al., 2017), (c) percentage of sand content (averaged throughout the soil profile) based on Soil Grids data, and (d) percentage of clay content (averaged throughout the soil profile) based on SoilGrids. The locations of the CosmOz network (Hawdon et al., 2014) stations 15 (Hamilton), 18 (Bishes) and 19 (Bennets) are indicated in figure (a).

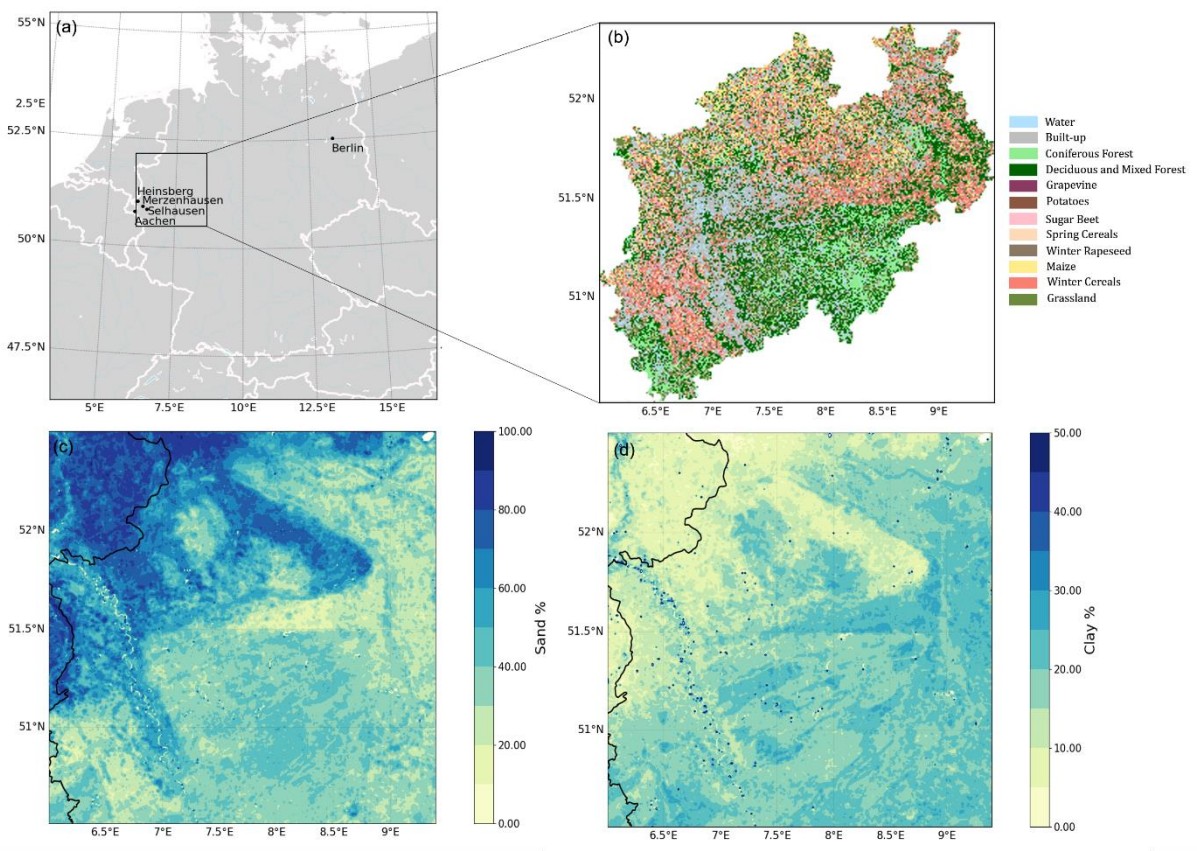

Figure 2: (a) DE-NRW simulation domain extent, (b) dominant land use type based on Griffiths et al. (2019), (c) percentage of sand content (averaged throughout the soil profile) based on Soil Grids data, and (d) percentage of clay content (averaged throughout the soil profile) based on Soil Grids. The locations of the COSMOS-Europe (Bogena et al., 2022) stations Merzenhausen, Heinsberg, Selhausen and Aachen are indicated in figure (a).

## 2.2    Agricultural statistics

The mild climate in Victoria is favourable to a range of winter crops, especially cereals (wheat, barley, oats), oilseeds (canola) and pulses (lentils, beans, chickpeas) contributing to Australia's total annual winter crop yield of ~5 million tons on average. Most of the crop production in Victoria is from the western and northern regions, expanding to high rainfall zones of southern Victoria. Wheat varieties represent the most commonly sown winter crop in Victoria with an average operated area of 1.3 million hectares (2015 to 2019 average) (ABARES, 2020). The production of summer crops such as grain sorghum, cotton or rice in Victoria is negligible with an average total production of 2,000 tons per year (2015 to 2021 average) (ABARES, 2020). The main cropping season in Victoria is from April to November. Regional average farming yield in the Victoria domain is highly influenced by seasonal rainfall patterns. In 2018, Victoria experienced substantial yield losses due to long dry spells and high temperatures after the first seasonal rainfalls, while record grain yields were recorded for the year 2020 (ABARES, 2020).

In the state of NRW, the most relevant cash crops are grain crops such as cereals (especially winter cereals) and corn, followed by canola, sugar beet and potatoes (BMEL, 2020, 2022). The main cropping season in Germany occurs during the spring and summer months until the beginning of fall from April to the end of October. The

European drought of 2018 lead to local yield losses, especially for the crops corn, potatoes, sugar beet and slightly for canola, as well as to unusually high spatial wheat yield variability within the region. The spatial variability was strongly related to soil type (IT.NRW, 2019; NRW Gov., 2020). Regions with clay rich soils that have high water holding capacities, saw unexpectedly high wheat yields in 2019, while regions dominated by less fertile sandy soils in the north-western part of the state experienced yield losses due to water deficits (NRW Gov., 2020). In general, the annual crop yield of the main cash crops varies more in Victoria than in NRW, where on a regional average, there is only small variation between the annual yields of the respective crops (Table 2, for a complete list of cropland area and production of major cash crops in Victoria and NRW, please see Table S1 and Table S2 in the supplementary material).

## 2.3    Land surface model

Land surface models such as CLM5 are essential tools for the study and prediction of terrestrial processes (e.g. energy, water and nutrient fluxes) and climate feedbacks in the terrestrial ecosystem, and are broadly applied in different scientific disciplines (e.g., Baatz et al., 2017; Lu et al., 2017; Chang et al., 2018; Han et al., 2018; Lawrence et al., 2018, 2019; Naz et al., 2019; Lombardozzi et al., 2020). In this study, the land surface model simulations were carried out with the latest version of the Community Earth System Model land component CLM5 which includes an adopted version of the prognostic crop module from the Agro-Ecosystem Integrated Biosphere Simulator (Kucharik and Brye, 2003; Lawrence et al., 2019b). CLM5 is forced by atmospheric states at a given time step and simulates the exchange of water, energy, carbon, and nitrogen between land and atmosphere as well as their storage and transport on the land surface and in the subsurface and the biomass and respective yield of crops upon harvest (Lawrence et al., 2019b; Lombardozzi et al., 2020). In CLM5, the plant hydraulic stress routine simulates water transport through the soil-root-stem-leaf system based on Darcy´s Law for porous media flow and adapts the vegetation water potential according to water supply with transpiration demand. Water stress for plants is based on leaf water potential which is used for the attenuation of photosynthesis in a transpiration loss function relative to maximum transpiration  (Lawrence et al., 2018). The leaf stomatal conductance and leaf photosynthesis are modelled for sunlit and shaded leaves separately based on the approaches after Medlyn et al. (2011), and Farquhar et al. (1980) for $C_3$ plants and Collatz et al. (1992) for $C_4$ plants (Lawrence et al., 2018) respectively. Adapted from Medlyn et al. (2011), the leaf stomatal resistance is calculated using the net leaf photosynthesis, the vapor pressure deficit and the $CO_2$ concentration at the leaf surface with plant-specific slope parameters (Lawrence et al., 2018).

With its biogeochemistry module, CLM5 is fully prognostic regarding crop phenology (e.g, grain yield, leaf area index, crop height) as well as carbon and nitrogen in the soil, vegetation and litter. The crop module includes a total of 78 plant and crop functional types, including an irrigated and unirrigated C3 crop, and crops such as winter wheat, spring wheat, canola temperate and tropical corn, temperate and tropical soybean, cotton, rice and sugarcane (Lawrence et al., 2018). Fertilization dynamics and annual fertilizer amounts in CLM5 depend on the crop functional types and vary spatially and yearly based on the land use and land cover change time series from the Land Use Model Intercomparison Project (Lawrence et al., 2019). Mineral fertilizer application starts during the leaf emergence phase of crop growth and continues for 20 days and manure nitrogen is applied at slower rates of 0.002 kg N m$^{-2}$ per year. For a more detailed description of the features and formulations of CLM5 the reader is referred to the technical description and latest literature (Lawrence et al., 2018, 2019a).

Here, we used a modified version of CLM5 that includes a winter cereal representation, an updated parameter set for several cash crops (winter wheat, sugar beet and potatoes), as well as a new subroutine that allows the simulation of cover cropping and a more flexible crop rotation (Boas et al., 2021). The modified CLM5 version led to significantly improved simulations of LAI, net ecosystem exchange, crop yield, and energy fluxes at several Central European sites (Boas et al., 2021).

## 2.4     Seasonal weather forecasts

In this study, we used long-range meteorological forecasts from ECMWF's fifth generation seasonal forecasting systems, SEAS5, which has been operational since November 2017 (Johnson et al., 2019). The SEAS5 forecasts are based on a coupled atmosphere-ocean model and provide forecasts of numerous meteorological variables at either 6-hourly or daily time step at a horizontal resolution of 1 degree. For the seasonal forecast, an ensemble of 51 members is initialised on the first day of a month and integrated for 7 months (Johnson et al., 2019). Furthermore, SEAS5 provides a set of retrospective seasonal hindcasts from 25 ensemble members for the years 1981 to 2016 that are used to calibrate and verify the forecasts compared to other datasets. While the whole period of hindcasts is used to verify the system, a subset from the years 1993 to 2016 is used in the calculation of forecast anomalies to avoid unreasonable affects from long-term climate trends on the forecast product (Johnson et al., 2019). A detailed description of the SEAS5 forecasting system and an overview of its performance is presented in Johnson et al. (2019). The SEAS5 forecasting product provides all variables needed to force CLM5 at daily or 6-hourly time step: accumulated daily precipitation amounts, daily shortwave and longwave radiation fluxes, wind speed, air temperature, dew point temperature and mean sea level pressure. In this study, we used the years 2017 to 2020 for our simulation experiments in accordance with the availability of the forecasting product.

We used different sets of SEAS5 forecast data, seasonal forecasts with 7-month lead-time and sub-seasonal forecasts with 3- and 4-month lead-time. Those variables available at only daily time step (incoming shortwave radiation and precipitation) were temporally disaggregated to a 6-hourly time step using the Meteorology Simulator (MetSim) (Bennett et al., 2020) to provide realistic information on atmospheric states. MetSim is based on algorithms from the Mountain Microclimate Simulation Model (MTCLIM) (Hungerford et al., 1989; Thornton and Running, 1999; Thornton et al., 2000; Bohn et al., 2013) and the Variable Infiltration Capacity (VIC) macroscale hydrologic model (Liang et al., 1994). MetSim can be used to either generate spatially distributed sub-daily time series of meteorological variables from a smaller number of input variables (daily minimum and maximum temperatures and elevation data), or to disaggregate meteorological data from a coarse temporal resolution to a finer one (Bennett et al., 2020).

 Besides the necessary meteorological input and calibration variables, MetSim also requires a grid description file that comprises information like spatial location (longitude and latitude), size of the grid cells and topography. Here, elevation data at a spatial resolution of 1 arc second from the ASTER Global Digital Elevation Model (ASTGTM) was used (NASA/METI/AIST/Japan Spacesystems And U.S./Japan ASTER Science Team, 2019).

The daily variables were disaggregated to sub-daily resolution. The total daily precipitation was split into 4 equal amounts of precipitation and then spread across the sub-daily time steps (6-hourly). Similar approaches were used for the NCEP dataset (Viovy, 2018) and in Hudiburg et al. (2013). Unfortunately, this deterministic approach cannot characterize the diurnal cycle of precipitation properly. The incoming shortwave radiation is disaggregated by multiplying the total daily shortwave by the fraction of radiation that is calculated by the solar geometry module

of MetSim. The solar geometry module within MetSim computes the daily potential radiation, day length and transmittance of the atmosphere based on the algorithms from MTCLIM (Thornton and Running, 1999). The influence of the temporal resolution of forcing data on simulation results and the quality of MetSim disaggregated data for the CLM5 model performance relative to hourly forcing data is illustrated and discussed for an example at point scale in the Appendix 6.1 and in the supplementary material.

## 2.5    Simulation experiments

We conducted simulation experiments using different sets of seasonal (up to 7-month lead time) and sub-seasonal (up to 4-month lead time) forecasts in order to assess a potential difference for the prediction of annual crop yield and general model system response for different forecast lead times. For the seasonal experiments (CLM-S), forecasts with a lead time of 7 months covering the main growing season (1st of April to 31st of October) were used. The seasonal simulations started on the 1st of April and continued for 7 months, until the end of October of the same year. The same time scale was used for the sub-seasonal experiments (CLM-SUB) that were forced with a combined set of forecasts with lead times of 3 and 4 months (from 1st of April until end of June and from the 1st of July until the 31st of October) (Figure 3). Seasonal and sub-seasonal experiments were conducted for the years 2017, 2018, 2019 and 2020 in order to assess the ability of the model to portray inter-annual differences in crop production, and for both domains. Furthermore, reference simulations (CLM-WFDE5) were conducted for the years 2017, 2018 and 2019 using the bias-adjusted global reanalysis dataset WFDE5 (Cucchi et al., 2020). The WFDE5 dataset has been generated from the ERA5 reanalysis product (Hersbach, 2016; Hersbach et al., 2020) using the WATCH Forcing Data (WFD) methodology (Cucchi et al., 2020). It is provided at 0.5° spatial resolution and at hourly time step for the period from 1979 to 2019.

An 850-year spin-up was performed prior to production runs for both domains in order to reach equilibrium conditions for soil carbon and nitrogen pools, soil water storage and other ecosystem variables. The global CRUNCEP atmospheric forcing dataset (Viovy, 2018) was used to force the spin-up simulations. The CRUNCEP dataset is a combination of the CRU TS3.2 0.5 x 0.5 degree monthly data covering the period 1901-2002 (Harris et al., 2014) and the NCEP reanalysis 2.5 x 2.5 degree 6-hourly data covering the period 1948-2016 (Kalnay et al., 1996).

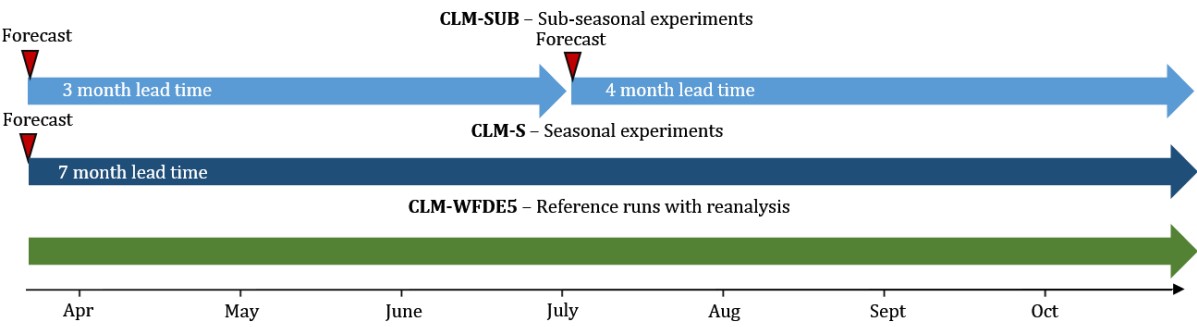

Figure 3: Schematic visualization of experimental simulation design.

In order to evaluate the quality of the simulation results we used the root mean square error (RMSE), the mean bias error (MBE) and the squared correlation coefficient ($R^2$) as statistical validation metrics:

$$RMSE = \sqrt{\frac{1}{n}\sum_{i=1}^{n}(X_i - y_i)^2}, \tag{1}$$

$$MBE = \frac{\sum_{i=1}^{n}(X_i - y_i)}{n}, \tag{2}$$

$$R^2 = 1 - \frac{\sum_{i=1}^{n}(y_i - X_i)^2}{\sum_{i=1}^{n}(y_i - \overline{y})^2}, \tag{3}$$

where $n$ the total number of time steps, $X_i$ and $y_i$ are the simulated and the observed values of a given variable at every time step $i$, and overbar represents mean value.

## 2.6    Validation data

For the validation of CLM5 simulated surface soil moisture, we compared simulation results with the Soil Moisture Active Passive (SMAP) mission Enhanced Level-3 radiometer soil moisture product (SMAP L3) (Entekhabi et al., 2016), and with the Soil Moisture CCI Combined dataset, version 05.2 (ESA-CCI), from the European Space Agency's (ESA) Soil Moisture Essential Climate Variable (ECV) Climate Change Initiative (CCI) project (Dorigo et al., 2017; Gruber et al., 2017, 2019). The global SMAP L3 product comprises soil moisture retrievals at both 6 a.m. and 6 p.m. at a spatial resolution of 9 km (Entekhabi et al., 2016). The ESA-CCI-SM combined product provides global daily volumetric soil moisture data at a spatial resolution of 0.25 degrees from 1978 to 2019. It was created by merging multiple scatterometer and radiometer soil moisture products (from AMI-WS, ASCAT; SMMR; SSM/I, TMI, AMSR-E, WindSat, AMSR2, SMOS and SMAP satellites) and covers the period from 1978 to 2019 (Dorigo et al., 2017; Gruber et al., 2017, 2019).

In addition, simulation results were compared to available SMC measurements from three cosmic-ray neutron sensor (CRNS) measurements. For AUS-VIC we used measurement data from the stations Hamilton (station 15), Bishes (station 18) and Bennets (station 19) that are part of the CosmOz network (Hawdon et al., 2014). For DE-NRW, CRNS measurements were obtained from the four COSMOS-Europe stations Selhausen, Merzenhausen, Aachen and Heinsberg (Bogena et al., 2022). For this comparison, simulation outputs from the closest grid point to the respective station were averaged with the weighting approach after Schrön et al. (2017).

In order to validate the regional LAI and ET simulation results, we used the latest MODIS satellite data product (MCD15A3H version 6). This includes the Combined Fraction of Photosynthetically Active Radiation (FPAR), and Leaf Area Index (LAI) product (Myneni et al., 2015), as well as the MODIS Evapotranspiration (ET)/Latent Heat Flux (LH) (MOD1A2 version 6) product (Running et al., 2017). The MODIS LAI product is a 4-day composite data set (combined acquisitions of both MODIS sensors located on NASA's Terra and Aqua satellites) on a 500 m global grid (Myneni et al., 2015). The MODIS ET product is an 8-day composite at a 500 m global resolution (Running et al., 2017). We compared simulated LAI and ET with monthly mean values from MODIS for cropland dominated land units throughout both domains.

An overview of acronyms used in this study is provided in the Appendix (Table A 1).

## 3    Results

### 3.1    Comparison of seasonal forecasts to recorded weather statistics

In a first step, the forecasts for both domains were compared to official weather statistics and trends.

In 2017, the weather in Victoria was generally slightly drier and warmer than average, however, the winter season was unusually cool, with record minimum temperatures in July and August (BOM, 2021). Annual rainfall was below average in most months, especially in June and July, which resulted in the driest winter season since 2006.

However, early growing season rainfall in April was more than 50% above average for large parts of the state (BOM, 2021). The year 2018 continued with drier and warmer than average weather, with the lowest annual rainfall amount since 2006 and an annual mean temperature of more than 1°C above average (reference period of 1980-2010) (BOM, 2021). In the southwest and south of Victoria, winter season rainfall was close to average, while below average rainfall amounts were recorded across the north and east of the state (BOM, 2021). Similar

to the previous years, 2019 was generally warmer and drier than average. Winter season rainfall showed high variability throughout the state, it was below average for large parts of Victoria in the north and east and above average in the south (BOM, 2021). The year 2020 continued with close to average rainfall and temperatures (BOM, 2021). The recorded weather pattern in Victoria is to a certain extent represented in the SEAS5 seasonal forecast data. The predicted state-wide average rainfall amount was highest for the fall and winter seasons (from April to

October) of 2020 and 2017, where recorded early season rainfall was 50 % above average, and lowest for 2018, where extremely low winter season rainfall was predicted. In NRW, the weather in 2017 was slightly warmer than the 30-year average with close to average rainfall. The year 2018 was characterized by an exceptional heat and drought wave during summer (Graf et al., 2020; DWD, 2021). Overall summer time rainfall in 2018 was below average which, in combination with high temperatures, led to exceptional drought conditions in NRW and most

of Europe that represent the largest annual soil moisture anomaly in the period 1979–2019 (Graf et al., 2020, and references therein). The same pattern, though less extreme, was observed in 2019, where a heat wave occurred during summer in combination with long dry spells. Total summer time rainfall was slightly below average. The year 2020 continued with above average summer time temperatures and below average rainfall making it the third too dry and too warm year in a row (DWD, 2021). The trend of the recorded weather patterns is to a certain extend

reflected in the SEAS5 forecasts for NRW. The predicted total rainfall over 7 months was lowest in the 2018 forecasts. The heat wave in 2018 is reflected in the forecasts in the predicted mean daily temperature that is more than 1°C higher than in 2017, 2019 and 2020.

### 3.2    Model performance with long-range forecasts

### 3.2.1    Soil moisture content

In general, the SMAP L3 data set depicts much stronger fluctuations in the soil moisture content (SMC) than the ESA-CCI product over both domains. Over the DE-NRW domain, SMAP L3 is drier in the early growing season and shows slightly wetter trend towards the end of the season compared to ESA-CCI (Figure 4). Large differences in SMC can be observed for the AUS-VIC domain, where SMAP L3 shows much higher magnitudes of SMC compared to ESA-CCI, in July-September in particular. Overall, the simulated SMC shows lower fluctuations for

the DE-NRW domain than for AUS-VIC. While the CLM5-simulated SMC for AUS-VIC corresponds better with the ESA-CCI product, for the DE-NRW domain CLM5-simulated SMC shows larger fluctuations and correlates better with the SMAP L3 product. For AUS-VIC, the CLM5 simulated SMC shows a wet trend towards the end of the winter season (August, September, October), especially for 2018 and 2019, compared to ESA-CCI (Figure 4). The reference runs CLM-WFDE5 generally correlated better with the ESA-CCI data ($R^2 > 0.8$) than the

seasonal and sub-seasonal runs ($R^2$ values between 0.2 and 0.64) for AUS-VIC (Table 1). Overall, the fluctuations

of the SMAP L3 product are not well represented in CLM5 simulated SMC over AUS-VIC. Both the forecast experiments and the reference simulations underestimated the SMC in comparison to SMAP L3 during the middle of the growing season for all years, while overestimating early and late growing season SMCs (Figure 4).

A different trend can be observed for the DE-NRW domain (Figure 4). While simulation results from CLM-S and CLM-SUB show a slight overestimation of the surface SMC in the beginning of the growing season (April to June) of 2017 and 2019 compared to the ESA-CCI product, a clear negative bias can be observed over summer and towards the end of the growing season (July to October) of 2017 and 2020 compared to ESA-CCI (Figure 4). This is also true for the CLM-WFDE5 run in 2018 and 2019. For 2017, CLM-WFDE5 overestimated the early season surface SMC but captured it relatively well towards the end of the season in reference to ESA-CCI (Figure 4). Compared to the SMAP L3 product, CLM5 overestimated early growing season SMC for all years, except for 2020, where a systematic underestimation of simulated SMC can be observed throughout the whole season. For the years 2018 and 2019, the SMAP L3 product seems to capture the recorded drought conditions in DE-NRW better compared to the ESA-CCI product, showing much lower SMCs. In the late growing season of 2019 (September and October), the SMAP L3 data as well as the ESA-CCI product show a prominent increase in SMC that is to a certain extend captured in the reference simulations, but not in the seasonal and sub-seasonal experiments. Overall, the CLM-WFDE5 simulations correlated better with both SMAP L3 and ESA-CCI ($R^2 > 0.54$ for all years) compared to forecast experiments ($R^2$ values between 0.12 and up to 0.42).

Only minor differences between the seasonal and sub-seasonal experiments can be observed for AUS-VIC, while for DE-NRW, the sub-seasonal experiment yielded lower mean soil moisture contents compared with the seasonal model runs in the late growing season, especially in August and September of 2017.

Because of the large differences between the two validation data sets ESA-CCI and SMAP L3 over AUS-VIC, we also compared the simulated SMC to available SMC measurements from three cosmic-ray neutron sensor (CRNS) stations (Station 15 – Hamilton, Station 18 – Bishes and Station 19 – Bennets) (Hawdon et al., 2014) for the years 2017 and 2018 (Figure A2). A relatively good correlation is reached for Hamilton during the early growing seasons of 2017 and 2018, while later in the season the SMC is underestimated. The simulated SMC is relatively high at the stations Bennets and Bishes (Figure A2) compared to CRNS data. We note that this comparison can only serve as an impression to give a tendency of model performance as simulation results and measurements may differ in soil types. For instance, Bishes and Bennets have a very sandy soil composition while in the SoilGrids data set the sand content is between 20 and 40 % (Figure 1). The station Hamilton is characterized by soils with a high water holding capacity which explains the high SMCs in the middle and towards the end of the wet season (Figure A2) which is not to that extent represented in the CLM5 simulations and underlying SoilGrids data. Single precipitation and/or flooding events that are reflected in the CRNS data are not represented in the forecasts and thus, naturally not captured in the simulation results. However, also the reference simulations were not able to represent these fluctuations (Figure A2). For DE-NRW, CLM5 simulations correspond better with SMAP L3 and show more fluctuations in day-to-day SMC. Here, the forecast experiments performed reasonably well in capturing the drought conditions with very low soil moisture contents throughout summer and fall in 2018 and 2019. We compared CRNS measurements from four stations within DE-NRW (Selhausen, Merzenhause, Aachen and Heinsberg) (Bogena et al., 2022) to the simulated SMC at the closest grid point respectively. The comparisons showed that the reference simulations forced with reanalysis generally produced higher SMCs than the forecast

simulations and corresponded better with CRNS measurement in terms of fluctuation intensity and magnitudes than SMCs from forecast simulations for single sites (Figure A3).

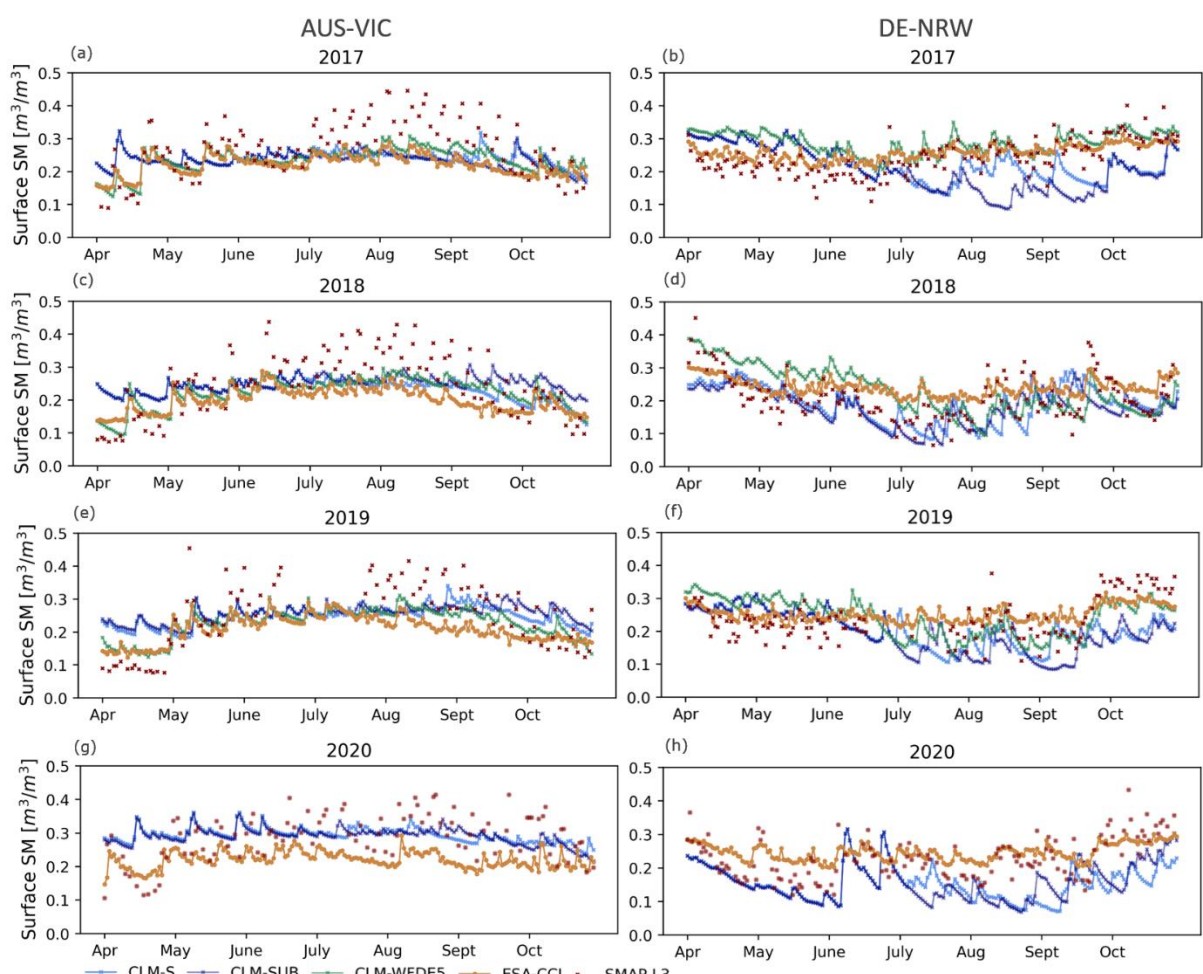

Figure 4: CLM-S, CLM-SUB and CLM-WFDE5 (for 2017, 2018 and 2019) simulated daily soil moisture content in the surface
layer (0 – 0.05 m) from April to October of 2017, 2018, 2019 and 2020 averaged over (left) the AUS-VIC domain and (right)
the DE-NRW domain, compared to the ESA-CCI surface soil moisture product and SMAP L3 data for the same time period
and domain respectively. Corresponding statistics (RMSE and bias) are listed in Table 1.

Table 1: RMSE, MBE and $R^2$ of CLM-S-, CLM-SUB- and CLM-WFDE5-simulated surface soil moisture [$m^3/m^3$] (0 – 0.05
450 m) from the 1$^{st}$ of April to the 31$^{st}$ of October of 2017, 2018, 2019 and 2020, compared to the ESA-CCI and SMAP L3 soil
moisture products, for the AUS-VIC and the DE-NRW domains.

| **AUS-VIC** | | | | | | | | | | | | |
|---|---|---|---|---|---|---|---|---|---|---|---|---|
| | 2017 | | | 2018 | | | 2019 | | | 2020 | | |
| | RMSE | MBE | $R^2$ | RMSE | MBE | $R^2$ | RMSE | MBE | $R^2$ | RMSE | MBE | $R^2$ |
| **SMAP L3** | | | | | | | | | | | | |
| CLM-S | 0.102 | 0.012 | 0.450 | 0.089 | 0.015 | 0.797 | 0.100 | 0.027 | 0.567 | 0.049 | 0.002 | 0.045 |
| CLM-SUB | 0.101 | 0.009 | 0.448 | 0.105 | 0.028 | 0.475 | 0.109 | 0.032 | 0.295 | 0.048 | 0.001 | 0.124 |
| CLM-WFDE5 | 0.094 | 0.009 | 0.629 | 0.086 | 0.012 | 0.708 | 0.085 | 0.014 | 0.751 | - | - | - |

| | RMSE | MBE | R² | RMSE | MBE | R² | RMSE | MBE | R² | RMSE | MBE | R² |
|---|---|---|---|---|---|---|---|---|---|---|---|---|
| **ESA-CCI** | | | | | | | | | | | | |
| CLM-S | 0.038 | 0.018 | 0.233 | 0.043 | 0.031 | 0.635 | 0.054 | 0.038 | 0.452 | 0.079 | 0.074 | 0.226 |
| CLM-SUB | 0.036 | 0.014 | 0.288 | 0.058 | 0.048 | 0.477 | 0.059 | 0.045 | 0.392 | 0.077 | 0.071 | 0.200 |
| CLM-WFDE5 | 0.022 | 0.014 | 0.886 | 0.033 | 0.023 | 0.846 | 0.029 | 0.019 | 0.881 | - | - | - |

| | | DE-NRW | | | | | | | | | | |
|---|---|---|---|---|---|---|---|---|---|---|---|---|
| | | 2017 | | | 2018 | | | 2019 | | | 2020 | |
| | RMSE | MBE | R² | RMSE | MBE | R² | RMSE | MBE | R² | RMSE | MBE | R² |
| **SMAP L3** | | | | | | | | | | | | |
| CLM-S | 0.068 | -0.011 | 0.186 | 0.056 | -0.010 | 0.420 | 0.065 | -0.016 | 0.190 | 0.079 | -0.047 | 0.123 |
| CLM-SUB | 0.083 | -0.023 | 0.259 | 0.058 | -0.016 | 0.412 | 0.071 | -0.024 | 0.199 | 0.075 | -0.044 | 0.404 |
| CLM-WFDE5 | 0.053 | 0.030 | 0.521 | 0.057 | 0.014 | 0.523 | 0.053 | 0.009 | 0.473 | - | - | - |
| **ESA-CCI** | | | | | | | | | | | | |
| CLM-S | 0.068 | -0.033 | 0.161 | 0.071 | -0.051 | 0.458 | 0.071 | -0.046 | 0.164 | 0.079 | -0.047 | 0.123 |
| CLM-SUB | 0.092 | -0.053 | 0.266 | 0.076 | -0.060 | 0.464 | 0.085 | -0.057 | 0.174 | 0.075 | -0.044 | 0.404 |
| CLM-WFDE5 | 0.040 | 0.029 | 0.583 | 0.058 | -0.010 | 0.621 | 0.049 | -0.011 | 0.548 | - | - | - |

### 3.2.2 Leaf area index and evapotranspiration

For AUS-VIC, the simulated LAI from seasonal and sub-seasonal experiments corresponds well with MODIS data, especially for the years 2017 and 2018. Only minor differences can be observed for 2017 and 2018 between the seasonal and sub-seasonal experiments and reference simulations. For 2019, CLM-S and CLM-SUB performed better than the reanalysis run which shows a systematic underestimation of LAI compared to MODIS throughout most of the cropping season. This is also reflected in CLM-WFDE5-simulated ET that is strongly underestimated for 2019 compared to MODIS. CLM5 simulation results for AUS-VIC generally show a systematic negative bias in simulated ET compared to MODIS data from April to August (Figure 5). The simulated inter-annual differences of LAI and ET are relatively small. For the DE-NRW domain, CLM5 overestimated the LAI compared to MODIS, in particular for the months June and July (Figure 6). For 2017 and 2018, CLM-WFDE5 resulted in very similar LAI values compared to CLM-S and CLM-SUB, while for 2019 the CLM-WFDE5-simulated LAI curve peaked later (highest LAI in August) compared to forecast simulations (highest LAI in July). Both CLM-S and CLM-SUB captured lower LAI magnitudes in August 2018 compared to the other years. In general, CLM-S and CLM-SUB show only minor differences in terms of LAI and ET. An exception is the year 2017, where CLM-SUB resulted in very similar LAI values compared to MODIS in September and October, while at the same time also resulting in a smaller underestimation of ET compared to CLM-S. Similar to the results for the other domain, the simulated inter-annual differences of LAI and ET are relatively small.

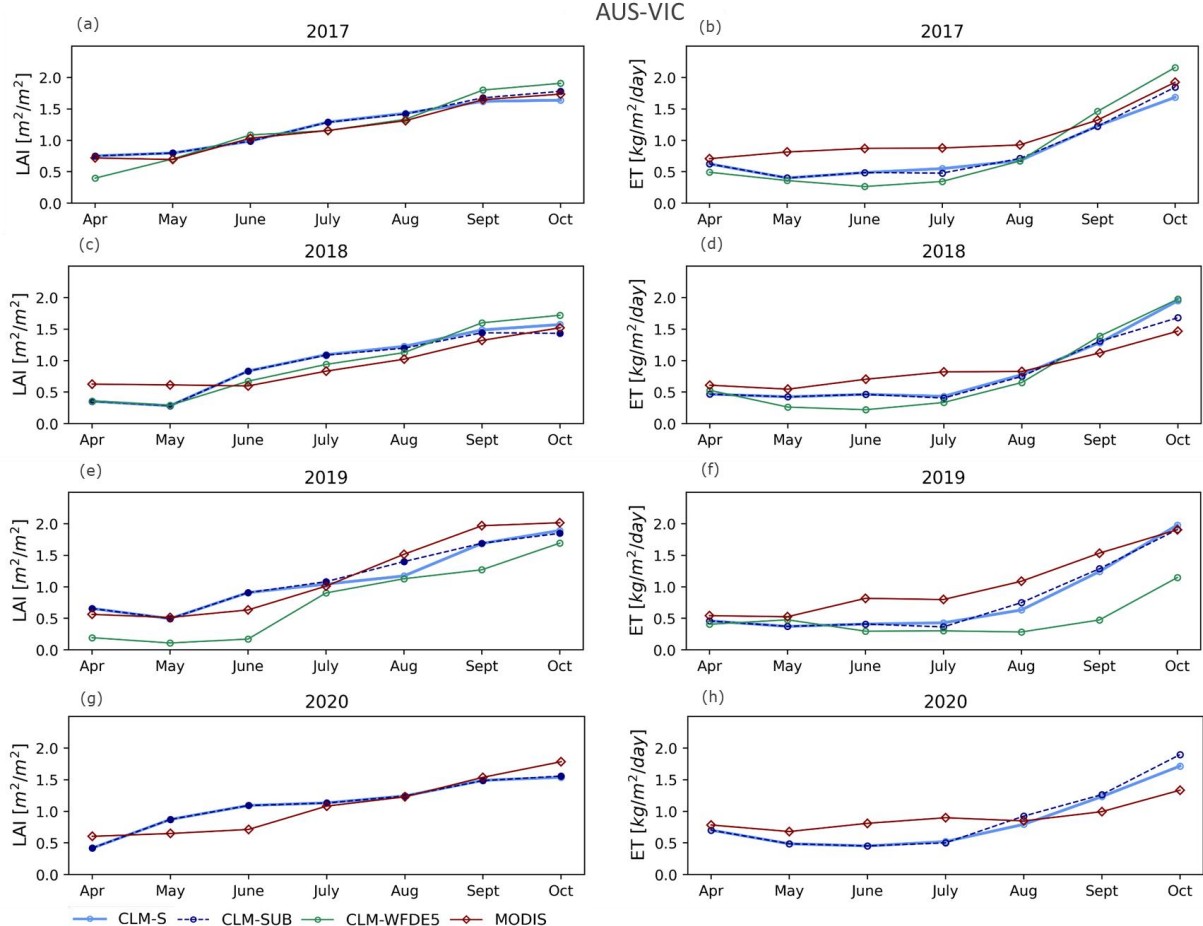

Figure 5: (left) Monthly mean LAI and (right) monthly mean ET derived from MODIS for April-October 2017-2020 compared to corresponding CLM seasonal (CLM-S) and sub-seasonal (CLM-SUB) simulation results, averaged over all land units with more than 70 % cropland within the AUS-VIC domain.

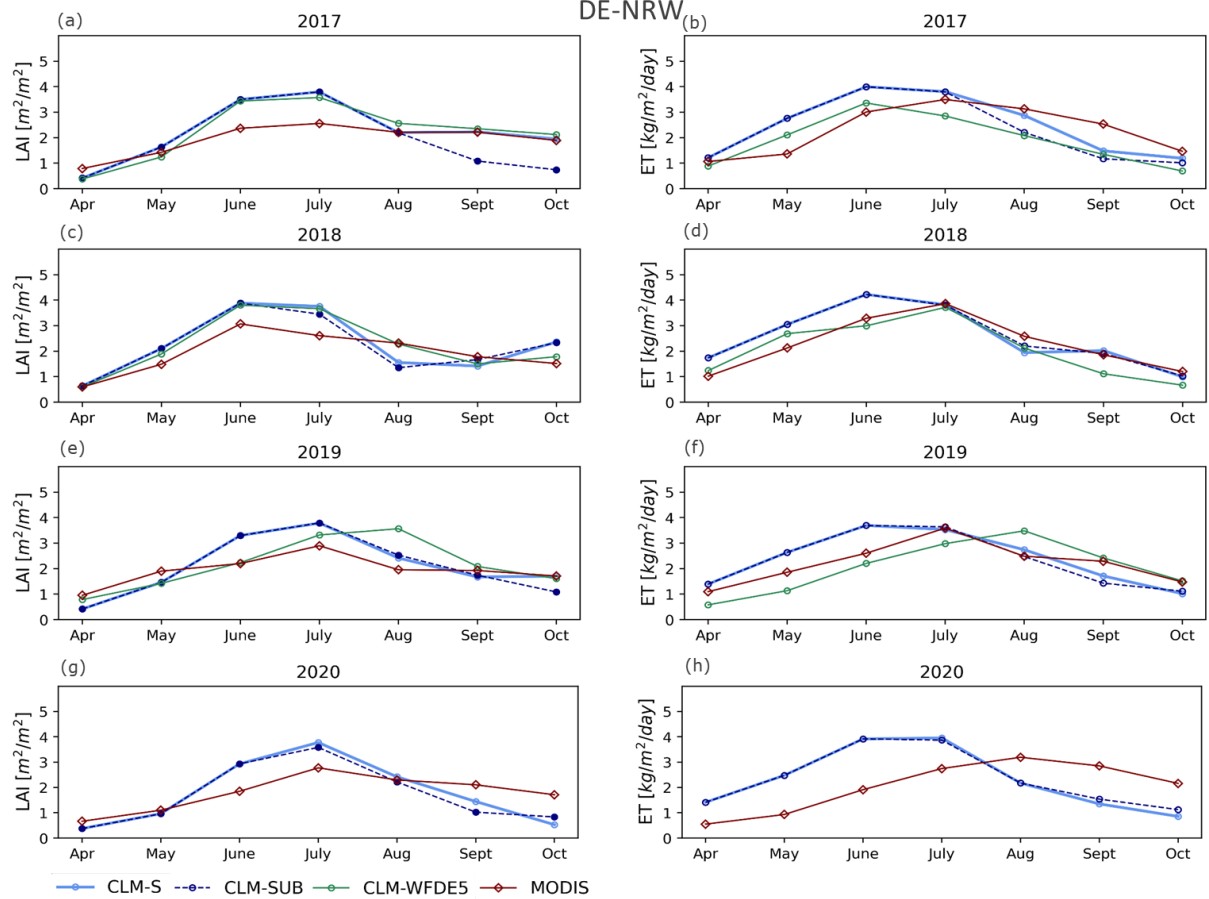

Figure 6: (left) Monthly mean LAI and (right) monthly mean ET derived from MODIS for April-October 2017- 2020 compared to corresponding CLM seasonal (CLM-S) and sub-seasonal (CLM-SUB) simulation results, averaged over all land units with more than 70 % cropland within the DE-NRW domain.

### 3.2.3 Regional crop yield predictions

CLM5 was able to reproduce the higher annual total crop yield for the DE-NRW domain compared to AUS-VIC (Figure 7, Table 2). For AUS-VIC, the simulations resulted in similar magnitudes of overall annual yield compared to statistics from the Australian Department of Agriculture, Water and Environment (ABARES) (Figure 7, Table 2). CLM-S and CLM-SUB systematically underestimated crop yield of all crops for the years 2017, 2019 and 2020, while overestimating crop yields for 2018 in comparison to official records. Still, the annual trends of 485 recorded crop yield were to a certain extend captured in the simulations. CLM-S and CLM-SUB showed the lowest yields in 2018 and slightly higher yields in 2017, 2019, and 2020, with 2020 being the most productive year in terms of total crop yield (Figure 7). Thus, for AUS-VIC, both the high-yield year of 2020 and the low-yield year of 2017 are well captured in the simulations. However, both the forecast experiments as well as the reference simulations resulted in a slightly lower overall yield for 2019 compared to 2017, which is the opposite in records. 490 CLM5 simulations generally showed lower inter-annual differences in crop yield compared to records. While the recorded annual crop yield varies up to 50 %, simulations resulted in differences of up to 17 % for the years 2017-2020. Inter-annual differences of the mean annual crop yield (averaged for the regarded crops) of up to 1.31 t/ha can be observed in the records, while crop yield simulated by CLM5 showed only differences of up to 0.30 t/ha in the forecast simulations (0.28 t/ha for CLM-SUB) and up to 0.24 t/ha in the reference simulations. In addition, we 495 observed a difference in the spatial distribution of crop productivity between the forecast experiments and

reference simulations. While in forecast experiments the highest crop productivity is simulated in the centre and north-eastern parts of the domain, the highest crop productivity in the reference simulations is located in the southern part of the domain closer to the coastline (Figure 8).

For the DE-NRW domain, the simulated crop yields are relatively close to recorded yields in terms of magnitudes 500 for all of the analysed cash crops wheat, corn, and canola (Figure 7, Table 2). The seasonal experiments were able to capture the high yield year of 2020 and the yield loss in 2018. In addition, the second and third highest yield years are captured in CLM-S and CLM-WFDE5 simulation results, but not in CLM-SUB simulation which had higher yields in 2019 than 2017 (Figure 7). CLM-S performed slightly better than CLM-SUB for all years in terms of total yields compared to records, except for 2018 where the CLM-SUB yield is lower and closer to records. 505 CLM5 simulations resulted in smaller inter-annual differences of the total annual crop yield with up to 6 % variation, compared to a recorded inter-annual difference of up to 15 % from 2017 to 2020. While inter-annual differences in crop yield up to 1.23 t/ha were observed in official records, CLM5 simulations resulted in smaller differences of up to 0.45 t/ha in CLM-S, 0.35 t/ha in CLM-SUB and 0.38 t/ha in reference simulations, on average for the regarded crops. There are no apparent spatial differences of simulated agricultural productivity between 510 the different experiments (Figure 9). Despite earlier enhancements to the model code and parameterization scheme (see Boas et al., 2021), the crop module of CLM5 does not include a proper representation of root crops. Here, we focus on the analysis of simulation results for wheat, corn and canola (Figure 7). An evaluation of simulation results for root crops can be found in the supplementary material (Chapter 4).

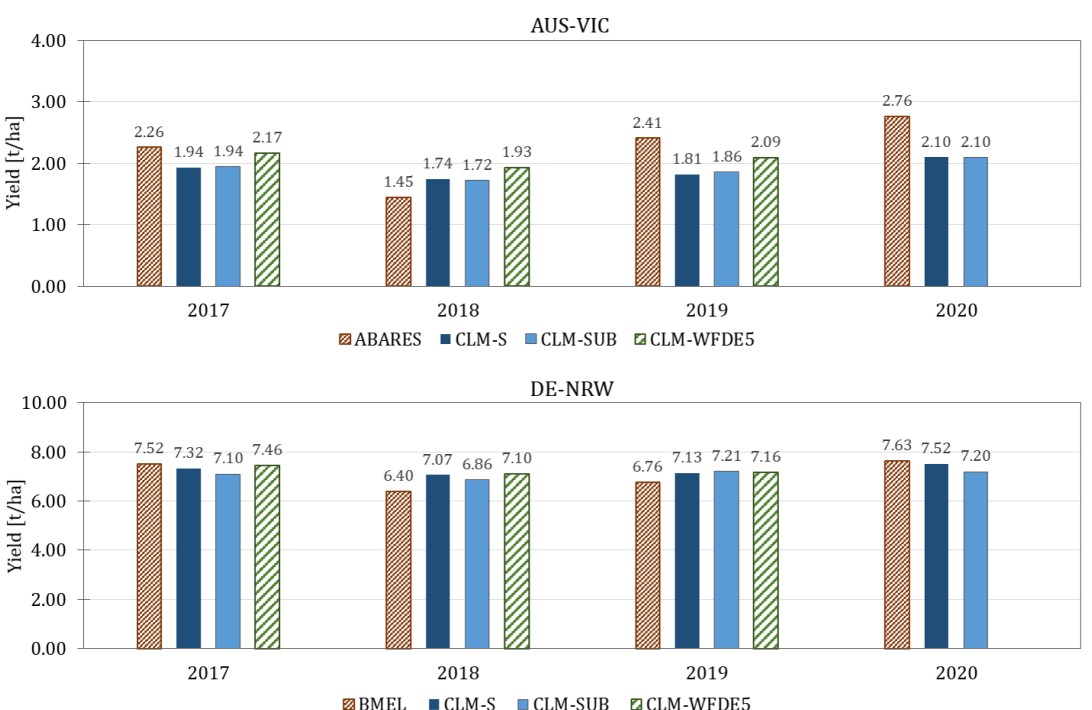

Figure 7: CLM-S-, CLM-SUB-, and CLM-WFDE5-simulated crop yield compared to corresponding official production records 515 (a) from ABARES (2020), averaged for all analysed winter crops (wheat, barley, and canola) within the AUS-VIC domain,

and (b) from BMEL (2020, 2022), averaged for all analysed crops (wheat, corn, and canola) within the DE-NRW domain, for the years 2017 to 2020. Corresponding data is listed in Table 2.

Table 2: Simulated crop yields [t/ha] for main cash crops with seasonal (CLM-S), sub-seasonal (CLM-SUB) and reanalysis (CLM-WFDE5) forcing data for the years 2017 to 2020, compared to official crop statistics from ABARES (2020) for the AUS-VIC domain and from BMEL (2020, 2022) for the DE-NRW domain. The lowest (italics) and highest yield (bold) amongst the respective years are indicated.

| | AUS-VIC | | | | | DE-NRW | | | |
| | 2017 | 2018 | 2019 | 2020 | | 2017 | 2018 | 2019 | 2020 |
|---|---|---|---|---|---|---|---|---|---|
| Wheat | | | | | Wheat | | | | |
| ABARES | 2.54 | *1.62* | 2.48 | **2.98** | BMEL | 7.92 | *7.91* | 8.14 | **8.66** |
| CLM-S | 2.15 | *2.05* | 2.15 | **2.23** | CLM-S | 7.96 | *7.59* | 7.61 | **8.19** |
| CLM-SUB | 2.15 | *2.03* | 2.19 | **2.23** | CLM-SUB | 7.57 | *7.24* | **7.76** | 7.67 |
| CLM-WFDE5 | 2.48 | 2.12 | 2.26 | - | CLM-WFDE5 | 8.04 | 7.41 | 7.67 | - |
| Barley | | | | | Corn | | | | |
| ABARES | 2.50 | *1.50* | 3.05 | **3.2** | BMEL | **10.74** | *7.80* | 8.44 | 10.49 |
| CLM-S | 2.46 | *2.15* | 2.17 | **2.47** | CLM-S | 9.27 | *9.12* | 9.27 | **9.68** |
| CLM-SUB | **2.47** | *2.12* | 2.20 | **2.47** | CLM-SUB | 9.21 | *9.06* | 9.34 | **9.29** |
| CLM-WFDE5 | 2.61 | 2.38 | 2.45 | - | CLM-WFDE5 | 9.72 | 9.31 | 9.26 | - |
| Canola | | | | | Canola | | | | |
| ABARES | 1.73 | *1.23* | 1.69 | **2.11** | BMEL | **3.90** | *3.48* | 3.69 | 3.74 |
| CLM-S | 1.20 | *1.03* | 1.13 | **1.35** | CLM-S | **4.73** | *4.49* | 4.52 | 4.69 |
| CLM-SUB | 1.21 | *1.02* | 1.18 | **1.35** | CLM-SUB | 4.53 | *4.28* | 4.54 | **4.63** |
| CLM-WFDE5 | 1.42 | 1.29 | 1.56 | - | CLM-WFDE5 | 4.62 | 4.59 | 4.46 | - |
| Average | | | | | Average | | | | |
| ABARES | 2.26 | *1.45* | 2.41 | **2.76** | BMEL | 7.52 | *6.40* | 6.76 | **7.63** |
| CLM-S | 1.94 | *1.74* | 1.81 | **2.02** | CLM-S | 7.32 | *7.07* | 7.13 | **7.52** |
| CLM-SUB | 1.94 | *1.72* | 1.86 | **2.02** | CLM-SUB | 7.10 | *6.86* | 7.21 | **7.20** |
| CLM-WFDE5 | 2.17 | 1.93 | 2.09 | - | CLM-WFDE5 | 7.46 | 7.08 | 7.16 | - |
| Ranking | | | | | | | | | |

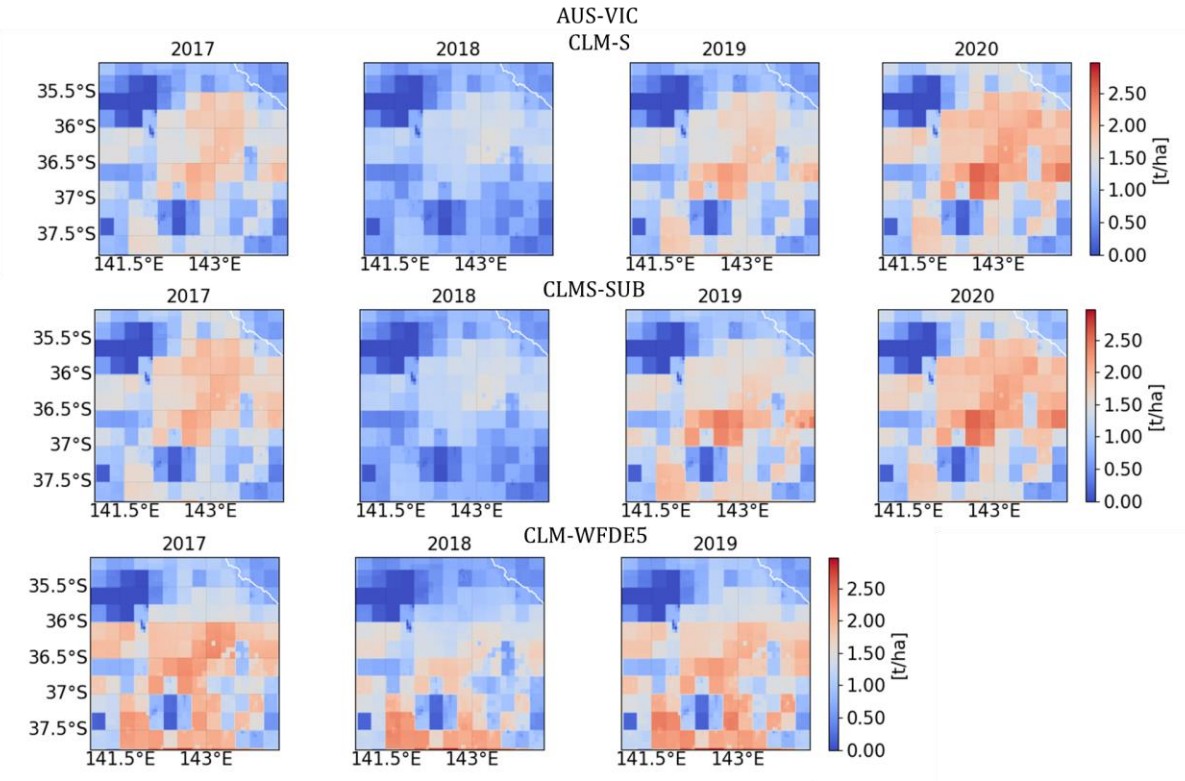

Figure 8: Spatial and inter-annual differences of the simulated annual crop yield (averaged) from (top) CLM-S, (middle) CLM-SUB, and (bottom) CLM-WFDE5 simulations throughout the AUS-VIC domain for the years 2017 to 2020.

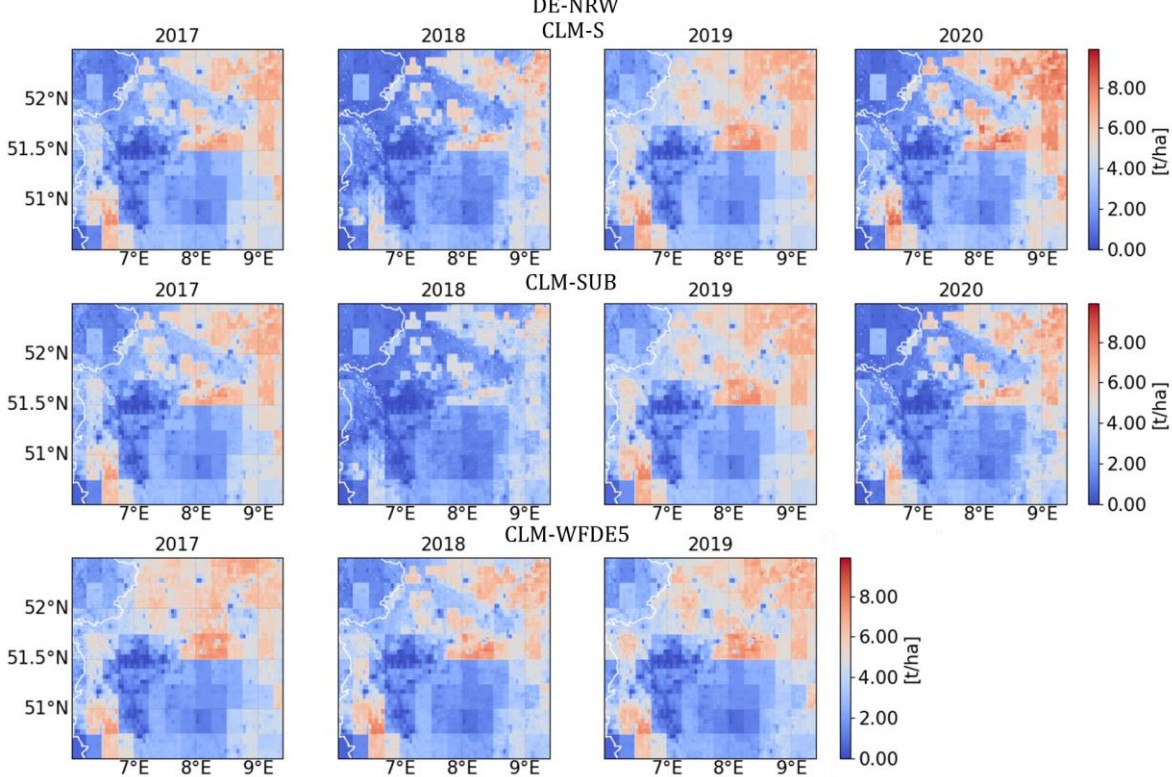

Figure 9: Spatial and inter-annual differences of the simulated annual crop yield (averaged) from (top) CLM-S, (middle) CLM-SUB, and (bottom) CLM-WFDE5 simulations throughout the DE-NRW domain for the years 2017 to 2020.

## 4        Discussion

Overall, annual crop yield predictions from the forecast experiments were close to results from the reference simulations, with a maximum difference between mean annual crop yield simulated with forecasts and with

reanalysis of 0.28 and 0.36 t/ha for AUS-VIC and DE-NRW, respectively. The forecast experiments were able to reproduce the recorded inter-annual trends of high yield year (2020) and low yield year (2018). In addition, the forecast experiments as well as the reference simulations were also able to reproduce the generally higher total values of annual crop yield for NRW compared to Victoria. The lower recorded crop yields in Victoria can be explained by less productive soils, limited water availability and different crop varieties. This is to a certain extent

also represented in CLM5 by a different parameterization and classification of northern and southern hemisphere crops. Furthermore, we used the CLM5 version and parameterization that was optimized for several European cropland sites and crops in an earlier study (Boas et al., 2021). The same study revealed a significant limitation of the default CLM5 phenology module and default crop parameterization to accurately represent European cropland sites, especially in terms of crop phenology (LAI magnitudes and seasonality), and grain yield (Boas et al., 2021).

Still, the inter-annual differences are lower in the CLM5 simulations compared to official yield statistics. On the one hand this could be due to the limited resolution and quality of the forecasts that predict general meteorological trends rather than realistic weather patterns (especially for precipitation). Although seasonal and sub-seasonal forecasts correctly predicted drier and hotter trends (e.g., for 2018), the 2018 drought was less pronounced in the forecast than in the observations. In addition, we observed a difference in the spatial pattern of crop productivity

over the AUS-VIC domain simulated with forecasts and reanalysis. Reference simulations resulted in a higher

crop production in the southern part of the domain than forecast experiments (Figure 8). This is related to the influence of near coastal precipitation events that are not well represented in the forecasts.

We found that the simulated LAI and ET corresponded reasonably well with data from MODIS in terms of magnitudes and fluctuations for the AUS-VIC domain, while for the DE-NRW simulations simulated LAI and ET

were larger than the observed values, in particular for the months of May, June and July. The better correlation between the simulated LAI and ET with observed values in the AUS-VIC domain, compared to DE-NRW, can be partly attributed to the larger paddock sizes and more homogeneous land cover in Victoria. The land cover in the state NRW is more diverse, with numerous urban areas and fallow lands between croplands that are not considered to the same extent by CLM5. Moreover, agricultural management practices and the variety of crop types and

cultivars are more diverse in DE-NRW, which is more challenging to represent accurately in simulations due to limitations in the input data and model structure. Several studies over European forest sites found lower absolute LAI values for MODIS compared to ground based measurements, and different seasonal dynamics that were partly explained by understory or herbal layer greening, and crypto- and microphytes in the understory that are not included in the measurements (e.g., Wang et al., 2005; Sprintsin et al., 2009). Earlier studies with CLM5 showed

relatively good correspondence between CLM5 simulated LAI and field measurements for several crops (Boas et al., 2021). For 2018, the seasonal experiments showed a relatively steep decline of LAI towards the end of the growing season that occurred earlier than for other years. The decline of LAI reflects the early simulation onset of harvest. The early harvest on a large part of the cropland in 2018 is closely linked to the recorded yield losses in NRW (Reinermann et al., 2019).

In general, the inter-annual differences in simulated LAI and ET were relatively low in the forecast experiments as well as in the reference simulations. This is also reflected in low inter-annual differences of simulated crop yields. The seasonal experiments were able to reproduce the generally higher inter-annual differences in crop yield throughout the AUS-VIC domain (up to 50 % in records and 17 % in simulated yields) compared to the DE-NRW domain (up to 15 % in records and 5 % in simulated yields). After weather conditions, regional agriculture and

crop yield is largely impacted by agricultural management decisions (e.g., on crop varieties, planting dates, irrigation, and fertilizer types and application techniques) as well as other environmental factors such as pests and crop damage from wildlife, which are not sufficiently well represented by CLM5. In addition, the crop module of CLM5 lacks parameterizations for most crop types and varieties and the fertilizer application routine is highly simplified. These deficiencies in the model structure led to considerable uncertainties in the crop phenology

simulated by CLM5.

Thus, the inter-annual variability in crop yield simulated by CLM5 is primarily influenced by the variability of model forcing data and soil moisture states, as it does not consider further anthropogenic or economic factors affecting crop yield, as discussed above. Consequently, the small inter-annual differences in simulated yield suggest that the CLM5 crop module has limited sensitivity to changes in climate conditions. Uncertainties in the

simulated annual crop productivity and its low inter-annual differences can be partly explained by the observed systematic biases of the simulated soil moisture content compared to satellite derived soil moisture products, i.e., ESA-CCI and SMAP L3, and CRNS measurements for both domains.

The reference simulations showed higher correlations between the simulated and observed surface soil moisture than the forecast experiments, which could be expected given the wrong timing of precipitation events in the

seasonal weather predictions, while still showing similar systematic differences compared to all products. Earlier

studies with CLM3.5 (e.g., Zhao et al., 2021; Hung et al., 2022) and CLM5 (e.g., Strebel et al., 2022) found pronounced discrepancies in CLM-simulated soil moisture contents and field measurements. In this context, data assimilation has proven to be a valuable technique to reproduce better soil moisture dynamics (Strebel et al., 2022). While the assimilation of soil moisture and groundwater level data into the Terrestrial Systems Modeling Platform (TSMP), which includes an earlier version of CLM (version 3.5), significantly improved simulated soil moisture properties and groundwater levels, it had only limited effects on resulting evapotranspiration (Hung et al., 2022). Whether a better representation of soil moisture within the model, i.e. through data assimilation, can significantly improve crop yield predictions with CLM5 remains to be evaluated.

The systematic uncertainties of the simulated soil moisture content as well as the low inter-annual differences in predicted crop yield and vegetation parameters (e.g., LAI and ET) show the need to improve the representation of these variables at the technical model level and to improve the model sensitivity to drought stress and other stressors (e.g., frost, pests, hail and wind). A sophisticated representation of crops and agricultural management in earth system models is essential in order to better assess the impact of climate change on yield in land surface models and specifically CLM5 (Lombardozzi et al., 2020). This includes e.g., the consideration of different types of fertilizers and application strategies, as well as a more detailed representation of root crops. It is crucial for the model to be sensitive enough to respond to changes in seasonality, drought stress and extreme events and realistically reflect these in resulting crop yields in order to study future yield scenarios. A better characterization of plant physiological and hydraulic properties, e.g. via plant trait information, is one suggestion for future model improvements. Studies over longer simulation periods are needed to consolidate whether this low inter-annual difference of CLM5 simulated crop yield is a systematic problem.

One major challenge in applying long-range forecast products in land surface models stems from the extensive pre-processing that is needed, including the temporal downscaling of certain meteorological variables (especially incoming shortwave radiation and precipitation). Simplifications in physical model formulations as well as uncertainties in the forcing data (e.g., due to coarse spatial and temporal resolution) may have impacted the simulated states. A more sophisticated temporal downscaling of precipitation, e.g. through machine learning techniques could help improve the applicability of forecasting products for model applications and improve the quality of model system responses. This becomes especially relevant when studying the impact of extreme events on agricultural productivity and other land surface processes. However, more sophisticated downscaling approaches often require further data sets that are not readily available. A clearer statement about the SEAS5 seasonal forecasting product regarding its overall quality for land surface modelling could be made once it is available for longer timescales. A performance analysis of available hindcasts over longer timescales and for further domains could provide a further systematic evaluation of the accuracy of the products in combination with CLM5. This could also benefit the creation of appropriate tools for end-users in order to increase the user-friendliness of the respective products. For future studies we additionally propose a benchmarking study of different forecasting products, e.g. from the German Weather Service (DWD), the National Centre for Environmental Prediction (NCEP), the CMCC Seasonal Prediction System, in combination with different land surface models like CLM5 that can point towards relative differences and limitations of each product in terms of applicability and overall skill. We believe that such a study, in addition to providing a better representation of the current state of the art in this field, will also benefit the exchange of knowledge at the interface between science and society.

## 5    Conclusion

The effects of climate change as well as the growing demand for food production entail vulnerability and challenges for regional agriculture and food security across all scales. Reliable high-resolution seasonal weather forecasting systems can provide important information for a multitude of weather-sensitive sectors when combined with a measurable model system response.


Here, we evaluated the quality and applicability of SEAS5 long-range meteorological forecasts in combination with CLM5 for two different regions. Our analysis illustrated that simulations forced with long-range forecasts were able to generate a model system response that was close to reference simulations which is an encouraging result for future studies. Both forecast and reanalysis forced models captured the inter-annual differences of yield

at least in sign (increase or decrease). The low and high yield seasons of 2018 and 2020 are clearly indicated for both simulated regions. The inter-annual differences of crop yield and other vegetation parameters (LAI and ET) were comparably low. Still, simulation results represented the higher inter-annual differencesin crop yield across the AUS-VIC domain compared to the DE-NRW domain. While general trends of soil moisture such as the drought in 2018 were reproduced in simulations, we found systematic over- and underestimations compared to different

validation data sets and site observations in both the forecast and the reference simulations that cannot be explained by uncertainties in the forecasting product alone. These systematic uncertainties of the simulated soil moisture and the low inter-annual differences of simulated vegetation parameters indicate the need for further technical model improvements.

Overall, this study provides a first impression on the utility and skill of the relatively new SEAS5 forecasting

system for land surface models and provides an evaluation of the CLM5 crop module potential for regional scale agricultural yield prediction in two different climate zones. Our evaluation and analysis of the CLM5 crop models performance set the stage for further model evaluation and improvements. A strong conclusion about the SEAS5 seasonal forecasting product regarding its overall quality for land surface modelling can be drawn once it is available for longer timescales. This research underlines the value of combining seasonal forecasts with land

surface models such as CLM5 or similar model applications (i.e. crop models).

## 6    Appendix

Table A 1: List of acronyms used in this study, their description and reference.

| Acronym | Description | Reference |
|---------|-------------|-----------|
| AUS-VIC | Simulation domain covering the large parts of the state Victoria, Australia | |
| CLM5 | The Community Land Model, version 5.0 | |
| CLM-S | Seasonal experiments forced with 7 month lead time forecasts | |
| CLM-SUB | Subseasonal experiments forced with a combined set of forecasts with lead times of 3 and 4 months | |
| CLM-WFDE5 | Reference simulations forced with reanalysis | |
| CRNS | Cosmic ray neutron sensor for measuring neutron count density, from which soil moisture is estimated | |
| CRUNCEP | Combined dataset of the CRU TS3.2 0.5 x 0.5 degree monthly data covering the period 1901-2002 (Harris et al., | Viovy (2018) |

| | 2014) and the NCEP reanalysis 2.5 x 2.5 degree 6-hourly data covering the period 1948-2016 | |
|---|---|---|
| DE-NRW | Simulation domain covering the state of North Rhine-Westphalia, Germany | |
| ECMWF | European Centre for Medium-Range Weather Forecasts | |
| ESA-CCI | Soil Moisture Climate Change Initiative Combined dataset from the European Space Agency's | Dorigo et al. (2017) |
| MetSim | Meteorology Simulator | Bennett et al. (2020). |
| MODIS | Satellite data product (MCD15A3H version 6) including the Leaf Area Index (LAI) product, as well as the MODIS Evapotranspiration (ET)/Latent Heat Flux (LH) (MOD1A2 version 6) product | Myneni et al., (2015); Running et al. (2017) |
| SEAS5 | Fifth generation seasonal forecasting system from ECMWF | Johnson et al. (2019) |
| SMC | Soil moisture content | |
| SMAP L3 | Soil Moisture Active Passive Level-3 soil moisture product | Entekhabi et al. (2016) |
| VLUIS | Victorian Land Use Information System | Morse-McNabb et al. (2017) |
| WFDE5 | Bias-adjusted global reanalysis dataset generated from the ERA5 reanalysis product (Hersbach, 2016; Hersbach et al., 2020) using the WATCH Forcing Data (WFD) methodology (Cucchi et al., 2020) | Cucchi et al. (2020) |

## 6.1 Effect of temporal forcing data resolution – a synthetic experiment

In order to analyse the overall effect of temporal forcing data resolution on model outputs and to assess the general need of temporal disaggregation from daily variables for CLM5 simulations, we performed a synthetic simulation experiment for a high resolution data set at point scale. We used a continuous measurement data set at hourly time step for 5 consecutive years from the cropland study site Selhausen (DE-RuS) located in the western part of Germany. Selhausen (50.86589°N, 6.44712°E) is part of the TERENO Rur Hydrological Observatory (Bogena at

al., 2018) and TERENO Eifel/Lower Rhine valley Observatory (Zacharias et al., 2011), as well as the Integrated Carbon Observation System (ICOS, 2020). Continuous measurements of meteorological variables and land-atmosphere exchange fluxes are available via the respective data portals (Kunkel et al., 2013; ICOS, 2020; TERENO, 2020). The original measurement data were first averaged to daily values and then temporally disaggregated to a 6-hourly time step using MetSim. Hence, simulations for a consecutive cycle of spring wheat

over 5 years (hypothetical) were conducted with the reference observation data at hourly time step, with daily averaged observations, and with the disaggregated 6-hourly forcing data set. A spin-up was conducted prior to this trial in order to balance ecosystem carbon and nitrogen pools, gross primary production and total water storage in the system (see Lawrence et al., 2018).

As expected the 6-hourly disaggregated data performed significantly better for all individual output variables than

the daily data, which performed poorly compared to the reference forcing. The effect is especially prominent for the soil water content and the surface runoff. Here, the 6-hourly disaggregated forcing was able to capture more realistic magnitudes of both soil moisture content and runoff, resulting in only a small wet bias compared to the reference forcing (see Table S5 in the supplementary material). The 6-hourly forcing resulted in a grain yield of 4.71 t/ha, which is relatively close to the grain yield with the hourly forcing of 4.9 t/ha, while the simulated grain

yield with the daily forcing of 4.12 t/ha is slightly lower. The soil moisture content (in the surface layers and the

root zone) plays an important role for the simulation of reasonable crop productivity, especially when trying to simulate inter-annual differences in crop yield and crop growth in response to e.g. drought conditions. However, in the given simulation example for the site DE-RuS, water availability in the root zone does not represent a main limiting factor for plant growth for the simulated years. This explains the small variations of simulated grain yield

and LAI with the different forcing data sets despite the profound differences in simulated soil water contents (Figure A1). The results from this trial underline the importance of an adequate temporal resolution for forcing data. For the seasonal weather forecast data, the temporal disaggregation of the product to an adequate temporal resolution is crucial in order to make the data suitable for comparable model applications. A more detailed overview of this experiment, as well as corresponding statistics are provided in the supplementary material.


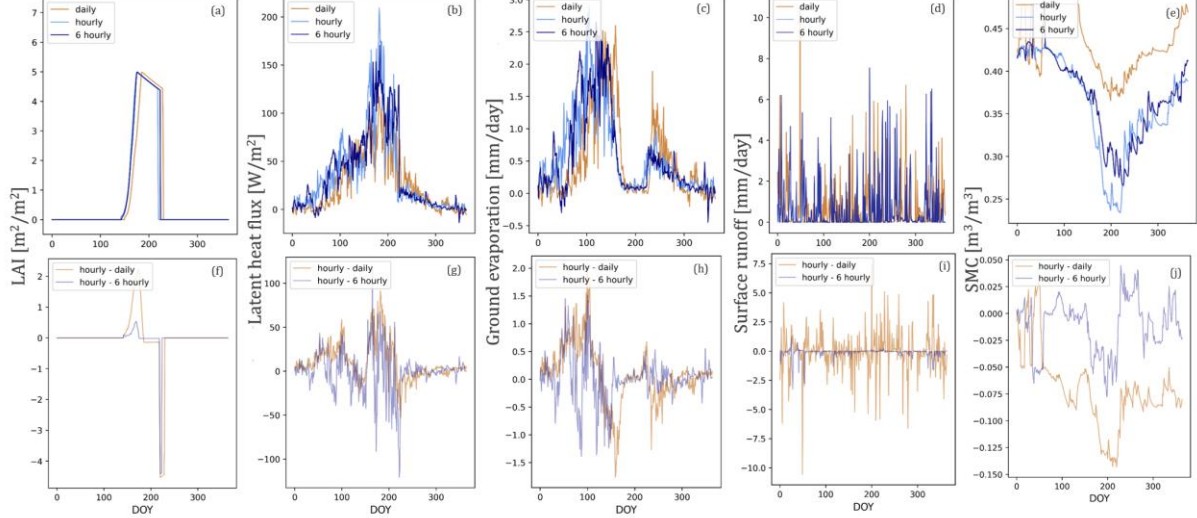

Figure A1: (Top) Comparison of simulation results for a cycle of (hypothetical) spring wheat cropping (averaged over 5 years) at DE-RuS with different temporal resolution of the forcing data: reference simulations forced with hourly observation data (light blue), daily averaged forcing data at 24 hour time step (orange) and disaggregated forcing data at 6-hourly resolution

(navy) for (a) LAI, (b) latent heat flux , (c) ground evaporation, (d) surface runoff, and (e) SMC (in an upper soil layer of 0.12 to 0.20 m depth). (Bottom) The difference of simulation results for each variable. Results from reference simulation forced

with hourly data minus the daily forcing (orange) and the 6 hourly disaggregated forcing (blue) respectively. Corresponding statistics are listed in Table S5 in the supplementary material.

## 6.2    Comparison with CRNS data

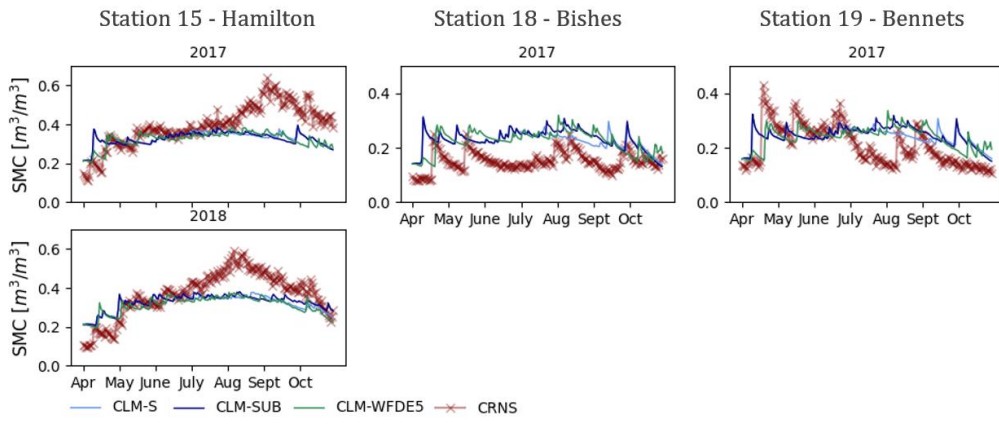

Figure A2: Comparison of CRNS data (level 4) from the stations 15 – Hamilton, 18 – Bishes and 19 – Bennets available via the CosmOz network (Hawdon et al., 2014) with simulated SMCs at the closest grid point for the years 2017 and 2018. Corresponding statistics can be found in the supplementary material.

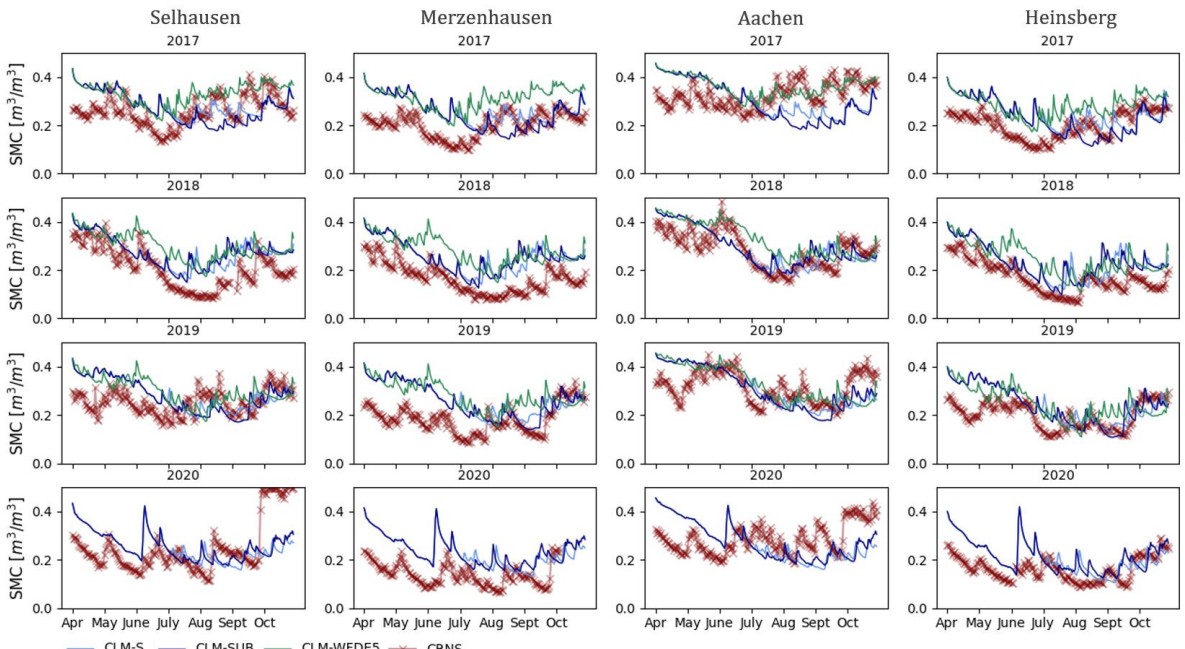

Figure A3: Comparison of CRNS data from the COSMOS-Europe sites Selhausen, Merzenhausen, Aachen and Heinsberg (Bogena et al., 2022) with simulated SMCs at the closest grid point for the years 2017-2020. Corresponding statistics can be found in the supplementary material.

## Code availability

The modified model version of CLM_ WW_CC that was used in this study is available online at
https://doi.org/10.5281/zenodo.3978092.

**Supplement link**

**Author contribution**

TB conceptualized and performed the simulation experiments, curated the data, analysed and visualized the simulation results and prepared the manuscript with contributions from all co-authors. HB, HJHF, DR, AW, and HV supervised the research, revised and edited the manuscript.

**Competing interests**

The authors declare the following competing interests: Harrie-Jan Hendricks Franssen is a part of the editorial board of Hydrology and Earth System Sciences.

**Acknowledgements**

The authors gratefully acknowledge the computing time granted on the supercomputer JUWLES by the Jülich Supercomputing Centre (JSC). This study is part of the Jülich-University of Melbourne Postgraduate Academy (JUMPA), an international research collaboration between the University of Melbourne, Australia, and the Research Centre Jülich, Germany.

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
