# Peer review of "Seasonal soil moisture and crop yield prediction with SEAS5 long-range meteorological forecasts in a land surface modelling approach"

_Hydrology and Earth System Sciences, 2023_

## Referee Comment (RC2)

Review of *Seasonal crop yield prediction with SEAS5 long-range meteorological forecasts in a land surface modelling approach*

The authors evaluate the use of seasonal (up to 7 months lead) and sub-seasonal (up to 4 months lead) weather forecasts for inputs into a land surface model to predict crop yields and other land surface features. Specifically they use the ECMWF SEAS5 forecasts with the Community Land Model (CLM) model to predict crop yields, soil moisture, leaf area index, and evapotranspiration. They evaluate forecast skill from 2017-2020 over two regions, one in Western Europe and another in Australia.  Overall, forecast crop yields were in-line with official statistics but were less successful in capturing inter-annual yield variations.  They also find systematic bias in soil moisture estimations when compared to two other observed Soil Moisture products( (SMAP L3 and ESA CCI).

This is a well written and executed study that is appropriate for HESS.  I have relatively few substantive concerns and several editorial recommendations and questions.

**Substantive Concerns**
My primary substantive concern is that this is a very short time period (2017-2020) and very limited spatial domain (1 in part of Germany another in part of Australia) from which to draw meaningful conclusions.  I understand the limitations on the temporal extent of the study (SEAS5 is a relatively new product that, as far as I know, has not been back cast). However I do not understand the limited spatial scope.  If the authors have agricultural stats for those regions in Germany and Australia, could they not conduct them for other regions? Or do national level comparisons using stats from either FAO, the USDA foreign ag-service (USDA-FAS), or another established source of crop yield/production data.  Given that the authors do find different levels of accuracy across the two regions, I would recommend adding a least one or two more regions or look at national level accuracy using FAO/USDA-FAS or another established source.  A comparison with national level statistics many countries in different biomes and production regimes (rainfed, irrigated, non-commercial, et cetera) would strengthen the paper.

Maybe expanding the spatial scope is not possible given the nature of the products in which case the authors can clarify this in their response to the reviewers. Regardless, in the paper the authors should clarify that forecast experiments over a relatively short period (and in a limited spatial domain) limit the applicability of the study for future use. This should not prevent publication, but the authors should clarify this and use analyses of long-term climatic and production trends in the study regions to elucidate what type of events (climate events, crop failures resulting from non-climate factors) this study might have missed and how that would impact using the results of this study either in the field for further research.  Again, expanding the spatial scope (and scales -ie national level) of the test sites could help alleviate some of the concerns that come from such a short temporal testing window.

**Editorial Recommendations**

- This paper uses A LOT of acronyms.  A simple table describing the main products and models (ie acronym, full name, resolution, source, et cetera) would really help the reader understanda and follow the results.
- Question and outline of paper not reached unitl page (around line 105)
- Line~126 Why not  include human components among dynamic nonlinear interactions- specifically, deliberate choices made by humans about resource allocation, investment, et cetera
    - Related on line 160, mention that farming decisions are a result of early season weather conditions- and for that matter seasonal and sub-seasonal weather forecasts similar to those used in this paper. To what extent to farmers in the study regions take advantage (and make decisions based on) regional outlooks that use similar forecasts? To what extent might this bias your results?
    - Lines 531-540 Why do you assume that inter-annual variability is driven only by climate factors (it may be, but you should clarify that assumption)?  Perhaps the model does not capture inter-annual variability precisely because it is driven by not climate factors (storage, global prices, costs of labor/land inputs, crop-disease, pests). In general the paper and discussion would be stronger if you account for the fact that grain yields are not a strictly bio-physical phenomenon (especially in the commercially farmed and irrigated regions you are studying).
- **Section 2.3**- This paper assumes a lot of familiarity with CLM5. I'm not sure if that holds with all readers of HESS (certainly not with this reviewer). Either in the main text or appendix, can you clarify what the key assumptions of the CLM5 are and how the parameters (ie production relative to input) are estimated or assumed.
- Line 255- the summary of weather patterns in 2017-2020 would be better captured in a figure
- Results—**Section 3.1** The title of the paper and much of the introduction focuses on Crop Yields. However, the first 2 pages of the results focus on soil moisture estimations. This is confusing to the reader. If these results are important (and it seems they are) both the title and intro material should also emphasize the soil moisture estimations as much as the crop yields.

---

## Author Comment (AC1)

**Replies to comments by Anonymous Referee #1, 06 April 2023**

In their article Boas et al. explore crop yield forecasts using seasonal and sub seasonal forecast models coupled to a land surface model that simulates crop yields. They explore the fidelity of their forecasts with respect to crop yields, and explore the source of muted year-to-year differences in yield levels in simulations relative to observations by evaluating the simulated LAI, ET, and soil moisture.

General comments:

Overall this research was well conducted and evaluated, but it could be communicated more effectively. The introduction contains information that belongs in the methods and the methods includes information that belongs in the results. The introduction in particular could be clarified to make the paper more accessible to the reader.

> Thank you very much for your constructive comments and suggestions. We have restructured large parts of our introduction and moved certain paragraphs from the introduction to methods, and from methods to the results section as indicated. Please find our detailed replies below.

**Replies to the list of specific comments by reviewer #1:**

**Abstract**:

L 23: please quantify what you mean by "very close to reference simulations results"

> We have clarified our results discussion throughout the manuscript and added several paragraphs in the results and discussion sections accordingly. Please also see our reply to the comments on the discussion below.

> (Line 22-25): We found that after preprocessing of the forecast products (i.e. temporal downscaling of precipitation and incoming shortwave radiation), the simulations forced with seasonal and sub-seasonal forecasts were able to provide a model output that was very close to the reference simulation results forced by reanalysis data (the mean annual crop yield showed a maximum difference of 0.28 and 0.36 t/ha for AUS-VIC and DE-NRW, respectively).

L26-28: I don't think "inter-annual variability" is the right phrase to use in the abstract. This would imply that you calculated a variance from a time series. You are instead describing the magnitude of the difference between two years relative to total yield levels. This portion of the abstract and the equivalent discussion in the results may be less confusing if you use the phrase "inter-annual differences" or describe the absolute change in yields between years (e.g. observations showed differences of up to XXXX tons/ha, while the model only simulated YYY) or similar.

> Thanks for pointing this out. We have decided to rephrase this accordingly throughout the manuscript.

> (Line 27-30): In addition, they also reproduced the generally higher inter-annual differences in crop yield across the AUS-VIC domain (approximately 50 % inter-annual differences in recorded yields and up to 17 % inter-annual differences in simulated yields) compared to the DE-NRW domain (approximately 15 % inter-annual differences in recorded yields and up to 5 % in simulated yields).

**Introduction:**

Although the vast majority of the paper is written clearly and effectively, the introduction is difficult to follow. The first and last paragraph in particular could be clarified to make the motivation and goals of the analysis more clear.

Mixing a discussion of the challenges of making climate forecasts with the challenges related to crop yield forecastss in the introduction can be difficult to follow for the reader. I suggest breaking the first paragraph into two paragraphs at around line 70. This would provide enough space to do a more thorough evaluation of crop yield forecasting literature as well. For example, in Australia see Wang et al. (2020) and Potgieter et al. (2022), as well as references therein, both of which are relevant to the present study.

Thanks for the suggestion. We gladly incorporated the suggested literature and added a paragraphs on crop yield modelling as follows:

(Line 108-116): Wang et al. (2020) investigated the impact of pre- and early-season El Niño Southern Oscillation (ENSO) related large scale climate signals on wheat yields in Australia. They found that these ENSO signals can have a significant impact on wheat yields in the Australian wheat belt and could explain up to 21% of the yield variation. In another study by Potgieter et al. (2022), the lead time and skill of Australian wheat yield forecasts using seasonal climate forecasts derived from a statistical ENSO-analogue system were compared with using a dynamic general circulation model (GCM). They found that ENSO-derived forecasts showed higher skills at a longer lead time (6 months), with a higher correlation coefficient of 0.48 compared to 0.37 for GCM forecasts, while GCM forecasts provided higher skill at shorter lead times (1-3 months) with a higher correlation coefficient of 0.44 compared to 0.35 for ENSO-analogue forecasts.

L75-82: discussion of MetSim and VIC belongs in the methods section, not the introduction

We agree and moved this paragraphs to the methods section:

(Line 281-287): MetSim is based on algorithms from the Mountain Microclimate Simulation Model (MTCLIM) (Hungerford et al., 1989; Thornton and Running, 1999; Thornton et al., 2000; Bohn et al., 2013) and the Variable Infiltration Capacity (VIC) macroscale hydrologic model (Liang et al., 1994). MetSim can be used to either generate spatially distributed sub-daily time series of meteorological variables from a smaller number of input variables (daily minimum and maximum temperatures and elevation data), or to disaggregate meteorological data from a coarse temporal resolution to a finer one (Bennett et al., 2020).

L115 - 137: Much of this information again belongs in the methods and makes the introduction difficult to follow. For example, specifying how you forced the MetSim model and how it was evaluated belongs in the methods.

Please see our reply to the previous comment.

L137-141: This is a nice and clear articulation of the study objectives. The paragraph leading up to this point is difficult to follow because these objectives were not outlined clearly. I would suggest moving these lines up to the beginning of the paragraph as it will help the reader to understand the study objectives before you go into detail about which models you use and why.

Thank you for your detailed suggestions for the introduction. We have restructured the text accordingly and moved the description of our study objectives towards the beginning of the section.

(Line 75-87): The major aim of this study was to evaluate the efficacy and applicability of this state-of-the-art forecasting product for physical and biogeochemical land surface responses and regional crop production in an ecosystem process model approach. To this end, we tested the combination of the Community Land Model version 5 (CLM5) (Lawrence et al., 2018; 2019) and seasonal forecasts from ECMWFs latest seasonal forecasting system SEAS5 (Johnson et al., 2019). Regional simulations were conducted for two domains with different climate regimes and agricultural characteristics, one covering the state of North Rhine-Westphalia in Germany (DE-NRW), and one the state of Victoria in Australia (AUS-VIC), using sub-seasonal and seasonal forecasts with different lead times as input. In our evaluations we focussed on (1) the model's sensitivity to seasonal changes in weather patterns and their effect on regional vegetation properties, e.g., leaf area index (LAI), evapotranspiration (ET), and crop yield; (2) the representation of the surface soil moisture content; and (3) the overall applicability and potential of seasonal weather forecasts for the prediction of regional agricultural production in model applications such as CLM5. In addition, we addressed the pre-processing steps required for the usage of the SEAS5 product in this model application and briefly discuss the importance of temporal downscaling.

**Materials and methods**

Section 2.4: There are periodically results sprinkled throughout your methods section. Please move these to the results section. For example, in L263-266 you discuss the performance of the forecasts relative to the observations. These belong in the results section as do all similar discussions of model performance and evaluation

We moved the whole paragraph with the comparison of seasonal forecasts and official weather statistics to the results section:

(Line 360-387): 3.1 Comparison of seasonal forecasts to recorded weather statistics

**Results:**

L383-4: "Only minor differences between the seasonal and sub-seasonal experiments can be observed for both domains". Is this true? It looks to be the case for AUS-VIC, but in the post-July period I see substantial differences In the DE-NRW domain between the sub seasonal and seasonal forecast runs.

> Thanks for pointing this out. We added a short comment on this:
>
> (Line 418-420): Only minor differences between the seasonal and sub-seasonal experiments can be observed for AUS-VIC, while for DE-NRW, the sub-seasonal experiment yielded lower mean soil moisture contents compared with the seasonal model runs in the late growing season, especially in August and September of 2017.

L454-456 and 465-467: As in the abstract, I think that the clarity of the results could be improved if you are more clear about what you are measuring here. The phrase "inter-annual differences" that you use is exceptionally clear, but then later you use the phrases "variation" and inter-annual variance, which could be confused with the variance of a time series. Consistently using "inter-annual differences" would clarify this confusion. Alternatively, you could describe differences using the absolute difference in yield levels rather than a percent.

> Thanks again for the suggestion. We understand how this wording could be confusing to the reader and thus have rephrased this throughout our manuscript, e.g.;
>
> (Line 570-587): In general, the inter-annual differences in simulated LAI and ET were relatively low in the forecast experiments as well as in the reference simulations. This is also reflected in low inter-annual differences of simulated crop yields. The seasonal experiments were able to reproduce the generally higher inter-annual differences in crop yield throughout the AUS-VIC domain (up to 50 % in records and 17 % in simulated yields) compared to the DE-NRW domain (up to 15 % in records and 5 % in simulated yields). After weather conditions, regional agriculture and crop yield is largely impacted by agricultural management decisions (e.g., on crop varieties, planting dates, irrigation, and fertilizer types and application techniques) as well as other environmental factors such as pests and crop damage from wildlife, which are not sufficiently well represented by CLM5. In addition, the crop module of CLM5 lacks parameterizations for most crop types and varieties and the fertilizer application routine is highly simplified. These deficiencies in the model structure lead to considerable uncertainties in the crop phenology simulated by CLM5.
>
> Thus, the inter-annual variability in crop yield simulated by the CLM5 is primarily influenced by the variability of model forcing data and soil moisture states, as it does not consider further anthropogenic or economic factors affecting crop yield, as discussed above. Consequently, the small inter-annual differences in simulated yield suggest that the CLM5 crop module has limited sensitivity to changes in climate conditions. Uncertainties in the simulated annual crop productivity and its low inter-annual differences can be partly explained by the observed systematic biases of the simulated soil moisture content compared to satellite derived soil moisture products, i.e., ESA-CCI and SMAP L3, and CRNS measurements for both domains.

**Discussion:**

L492: Perhaps use language that is more precise than saying that the forecasts were "very close" to the results of the reference simulations. The phrase does not give a lot of guidance to the reader and what is considered "very close" is subjective and often application dependent. You could, for example, describe which aspects of the simulations were well simulated (inter-annual yield differences, overall yield levels, etc).

> Thanks for pointing this out. We added results in form of total yield differences (t/ha) in the results section and discussion:
>
> (Line 492-494): Inter-annual differences of the mean annual crop yield (averaged for the regarded crops) of up to 1.31 t/ha can be observed in the records, while crop yield simulated by CLM5 showed only differences of up to 0.30 t/ha in the forecast simulations (0.28 t/ha for CLM-SUB) and up to 0.24 t/ha in the reference simulations.

(Line 506-509): While inter-annual differences in crop yield up to 1.23 t/ha were observed in official records, CLM5 simulations resulted in smaller differences of up to 0.45 t/ha in CLM-S, 0.35 t/ha in CLM-SUB and 0.38 t/ha in reference simulations, on average for the regarded crops.

(Line 533-535): Overall, annual crop yield predictions from the forecast experiments were close to results from the reference simulations, with a maximum difference between the mean annual crop yield simulated with forecasts and with reanalyses of 0.28 and 0.36 t/ha for AUS-VIC and DE-NRW, respectively.

L531-533: See above discussion about terminology with respect to "inter-annual yield differences" vs "inter-annual variability"

Please see previous comments.

---

## Author Comment (AC2)

**Replies to comments by Anonymous Referee #2, 11 April 2023**

The authors evaluate the use of seasonal (up to 7 months lead) and sub-seasonal (up to 4 months lead) weather forecasts for inputs into a land surface model to predict crop yields and other land surface features. Specifically they use the ECMWF SEAS5 forecasts with the Community Land Model (CLM) model to predict crop yields, soil moisture, leaf area index, and evapotranspiration. They evaluate forecast skill from 2017-2020 over two regions, one in Western Europe and another in Australia. Overall, forecast crop yields were in-line with official statistics but were less successful in capturing inter-annual yield variations. They also find systematic bias in soil moisture estimations when compared to two other observed Soil Moisture products ( (SMAP L3 and ESA CCI).

This is a well written and executed study that is appropriate for HESS. I have relatively few substantive concerns and several editorial recommendations and questions.

> Thank you very much for your constructive suggestions. We have added several paragraphs in the introduction and discussion section in response to your comments. Please find our detailed replies below.

**Substantive Concerns**

My primary substantive concern is that this is a very short time period (2017-2020) and very limited spatial domain (1 in part of Germany another in part of Australia) from which to draw meaningful conclusions. I understand the limitations on the temporal extent of the study (SEAS5 is a relatively new product that, as far as I know, has not been back cast). However I do not understand the limited spatial scope. If the authors have agricultural stats for those regions in Germany and Australia, could they not conduct them for other regions? Or do national level comparisons using stats from either FAO, the USDA foreign ag-service (USDA-FAS), or another established source of crop yield/production data. Given that the authors do find different levels of accuracy across the two regions, I would recommend adding a least one or two more regions or look at national level accuracy using FAO/USDA-FAS or another established source. A comparison with national level statistics many countries in different biomes and production regimes (rainfed, irrigated, non-commercial, et cetera) would strengthen the paper.

Maybe expanding the spatial scope is not possible given the nature of the products in which case the authors can clarify this in their response to the reviewers. Regardless, in the paper the authors should clarify that forecast experiments over a relatively short period (and in a limited spatial domain) limit the applicability of the study for future use. This should not prevent publication, but the authors should clarify this and use analyses of long-term climatic and production trends in the study regions to elucidate what type of events (climate events, crop failures resulting from non-climate factors) this study might have missed and how that would impact using the results of this study either in the field for further research. Again, expanding the spatial scope (and scales -ie national level) of the test sites could help alleviate some of the concerns that come from such a short temporal testing window.

> Thank you for pointing this out. We strongly agree that further studies for other biomes and regions would be very interesting and evaluations over longer timescales are needed to provide a more conclusive overview of the forecasts skill. For this first feasibility study we limited our analysis to two domains with high resolution input data on land cover and land use and meaningful validation data that was available to us. In addition, we selected regions where agricultural land is largely rain-fed to minimize the impact of irrigation. An extension of the Australian domain would include regions of supplementary irrigation. This is important because the modelling of irrigation with CLM5 is subject to high uncertainties and thus would have affected our results. We also want to indicate that setting-up a large high-resolution domain in CLM5 is very time intensive and computationally expensive and, at national or global scale would likely need an intensive parameter calibration for the dominant crop types before generating plausible results.
>
> To address some of the abovementioned concerns, we have added a more thorough introduction on the challenges of crop modelling and added a more detailed paragraph discussing the other factors that impact crop productivity apart from weather conditions and that are missed in the land surface simulations.
>
> In addition, we explicitly mention in our discussion that future studies need to consider longer time periods and other domains, coupled with a variety of forecasting products and modelling applications.
>
> (Line 574-580): After weather conditions, regional agriculture and crop yield is largely impacted by agricultural management decisions (e.g., on crop varieties, planting dates, irrigation, and fertilizer types and application techniques) as well as other environmental factors such as pests and crop damage from wildlife, which are not sufficiently well represented by CLM5. In addition, the crop module of CLM5 lacks

parameterizations for most crop types and varieties and the fertilizer application routine is highly simplified. These deficiencies in the model structure lead to considerable uncertainties in the crop phenology simulated by CLM5.

(Line 619-630): A clearer statement about the SEAS5 seasonal forecasting product regarding its overall quality for land surface modelling could be made once it is available for longer timescales. A performance analysis of available hindcasts over longer timescales and for further domains could provide a further systematic evaluation of the accuracy of the products in combination with CLM5. This could also benefit the creation of appropriate tools for end-users in order to increase the user-friendliness of the respective products. For future studies we additionally propose a benchmarking study of different forecasting products, e.g. from the German Weather Service (DWD), the National Centre for Environmental Prediction (NCEP), the CMCC Seasonal Prediction System, in combination with different land surface models like CLM5 that can point towards relative differences and limitations of each product in terms of applicability and overall skill. We believe that such a study, in addition to providing a better representation of the current state of the art in this field, will also benefit the exchange of knowledge at the interface between science and society.

**Editorial Recommendations**

This paper uses A LOT of acronyms. A simple table describing the main products and models (ie acronym, full name, resolution, source, et cetera) would really help the reader understand and follow the results.

We added a table to the appendix to provide a better overview.

(Line 657): Table A 1: List of acronyms used in this study, their description and reference.

| Acronym | Description | Reference |
|---|---|---|
| AUS-VIC | Simulation domain covering the large parts of the state Victoria, Australia | |
| CLM5 | The Community Land Model, version 5.0 | |
| CLM-S | Seasonal experiments forced with 7 month lead time forecasts | |
| CLM-SUB | Subseasonal experiments forced with a combined set of forecasts with lead times of 3 and 4 months | |
| CLM-WFDE5 | Reference simulations forced with reanalysis | |
| CRNS | Cosmic ray neutron sensor for measuring neutron count density, from which soil moisture is estimated | |
| CRUNCEP | Combined dataset of the CRU TS3.2 0.5 x 0.5 degree monthly data covering the period 1901-2002 (Harris et al., 2014) and the NCEP reanalysis 2.5 x 2.5 degree 6-hourly data covering the period 1948-2016 | Viovy (2018) |
| DE-NRW | Simulation domain covering the state of North Rhine-Westphalia, Germany | |
| ECMWF | European Centre for Medium-Range Weather Forecasts | |
| ESA-CCI | Soil Moisture Climate Change Initiative Combined dataset from the European Space Agency's | Dorigo et al. (2017) |
| MetSim | Meteorology Simulator | Bennett et al. (2020). |
| MODIS | Satellite data product (MCD15A3H version 6) including the Leaf Area Index (LAI) product, as well as the MODIS Evapotranspiration (ET)/Latent Heat Flux (LH) (MOD1A2 version 6) product | Myneni et al., (2015); Running et al. (2017) |
| SEAS5 | Fifth generation seasonal forecasting system from ECMWF | Johnson et al. (2019) |
| SMC | Soil moisture content | |
| SMAP L3 | Soil Moisture Active Passive Level-3 soil moisture product | Entekhabi et al. (2016) |
| VLUIS | Victorian Land Use Information System | Morse-McNabb et al. (2017) |
| WFDE5 | Bias-adjusted global reanalysis dataset generated from the ERA5 reanalysis product (Hersbach, 2016; Hersbach et al., 2020) using the WATCH Forcing Data (WFD) methodology (Cucchi et al., 2020) | Cucchi et al. (2020) |

Question and outline of paper not reached unitl page (around line 105)

> We have restructured large parts of our introduction to better communicate the objectives and aims of this study.

Line~126 Why not include human components among dynamic nonlinear interactions- specifically, deliberate choices made by humans about resource allocation, investment, et cetera

> We have added more background on other factors that impact state-wide crop yield throughout the manuscript. Please also see the replies to the next comments.

Related on line 160, mention that farming decisions are a result of early season weather conditions- and for that matter seasonal and sub-seasonal weather forecasts similar to those used in this paper. To what extent to farmers in the study regions take advantage (and make decisions based on) regional outlooks that use similar forecasts? To what extent might this bias your results?

> We have added the following to the introduction:
>
> (Line 137-154): Despite their potential economic value for agricultural production systems, the quantitative adoption of seasonal climate forecasts from farmers is low, both in Victoria and NRW (e.g., Parton et al., 2019). The Australian Bureau of Meteorology attributed this to insufficient data and evidence about their value and conducted a series of studies of the potential value of a forecast based on a particular production system and for specific regions and timescales (Hansen, 2002; Hansen et al., 2006). Furthermore, the challenges highlighted above have hindered a widespread application of such long-range forecasts in agriculture, particularly for larger (not site-specific) scales (Coelho and Costa, 2010; Calanca et al., 2011). The lack of user-friendly tools and services that can provide crop-specific yield information based on seasonal forecasts and account for other economic factors (e.g., political choices, outlook for crop markets, etc.) is another constraint.

Lines 531-540 Why do you assume that inter-annual variability is driven only by climate factors (it may be, but you should clarify that assumption)? Perhaps the model does not capture inter-annual variability precisely because it is driven by not climate factors (storage, global prices, costs of labor/land inputs, crop-disease, pests). In general the paper and discussion would be stronger if you account for the fact that grain yields are not a strictly bio-physical phenomenon (especially in the commercially farmed and irrigated regions you are studying).

> In this sentence we referred to the simulated model yields and rephrased the sentences for better clarity. We agree that the influence of agricultural management decisions is also a major factor influencing regional yield records and to a large extent neglected in the model formulation. We very briefly refer to this when comparing the simulation results for the two domains. However, we agree that this should be further detailed and added a paragraph in the discussion:
>
> (Line 555-561): The better correlation between the simulated LAI and ET with observed values in the AUS-VIC domain, compared to DE-NRW, can be partly attributed to the larger paddock sizes and more homogeneous land cover in Victoria. The land cover in the state NRW is more diverse, with numerous urban areas and fallow lands between croplands that are not considered to the same extent by CLM5. Moreover, agricultural management practices and the variety of crop types and cultivars are more diverse in DE-NRW, which is more challenging to represent accurately in simulations due to limitations in the input data and model structure.
>
> (Line 570-587): In general, the inter-annual differences in simulated LAI and ET were relatively low in the forecast experiments as well as in the reference simulations. This is also reflected in low inter-annual differences of simulated crop yields. The seasonal experiments were able to reproduce the generally higher inter-annual differences in crop yield throughout the AUS-VIC domain (up to 50 % in records and 17 % in simulated yields) compared to the DE-NRW domain (up to 15 % in records and 5 % in simulated yields). Next to weather conditions, the regional agriculture and crop yield is largely impacted by agricultural management decisions, e.g., on crop varieties, planting dates, fertilizer types and application techniques, irrigation as well as other environmental factors such as pests and crop damage from wildlife, which are not sufficiently represented by CLM5. In addition, the crop module of CLM5 lacks parameterizations for

most crop types and varieties and the fertilizer application routine is highly simplified. These deficiencies in the model structure led to considerable uncertainties in the crop phenology simulated by CLM5.

Thus, the inter-annual variability in crop yield simulated by CLM5 is primarily influenced by the variability of model forcing data and soil moisture states, as it does not consider further anthropogenic or economic factors affecting crop yield, as discussed above. Consequently, the small inter-annual differences in simulated yield suggest that the CLM5 crop module has limited sensitivity to changes in climate conditions. Uncertainties in the simulated annual crop productivity and its low inter-annual differences can be partly explained by the observed systematic biases of the simulated soil moisture content compared to satellite derived soil moisture products, i.e., ESA-CCI and SMAP L3, and CRNS measurements for both domains.

**Section 2.3-** This paper assumes a lot of familiarity with CLM5. I'm not sure if that holds with all readers of HESS (certainly not with this reviewer). Either in the main text or appendix, can you clarify what the key assumptions of the CLM5 are and how the parameters (ie production relative to input) are estimated or assumed.

Thanks for pointing this out. We have added a short paragraph in the methods section on the crop module of CLM5. For more details on the complex formulations within the model, the reader is referred to the technical documentation and most recent literature on model configurations.

(Line 238-257): In CLM5, the plant hydraulic stress routine simulates water transport through the soil-root-stem-leaf system based on Darcy´s Law for porous media flow and adapts the vegetation water potential according to water supply with transpiration demand. Water stress for plants is based on leaf water potential which is used for the attenuation of photosynthesis in a transpiration loss function relative to maximum transpiration (Lawrence et al., 2018). The leaf stomatal conductance and leaf photosynthesis are modelled for sunlit and shaded leaves separately based on the approaches after Medlyn et al. (2011), and Farquhar et al. (1980) for $C_3$ plants and Collatz et al. (1992) for $C_4$ plants (Lawrence et al., 2018) respectively. Adapted from Medlyn et al. (2011), the leaf stomatal resistance is calculated using the net leaf photosynthesis, the vapor pressure deficit and the $CO_2$ concentration at the leaf surface with plant-specific slope parameters (Lawrence et al., 2018).

With its biogeochemistry module, CLM5 is fully prognostic regarding crop phenology (e,g, grain yield, leaf area index, crop height) as well as carbon and nitrogen in the soil, vegetation and litter. The crop module includes a total of 78 plant and crop functional types, including an irrigated and unirrigated C3 crop, and crops such as winter wheat, spring wheat, canola temperate and tropical corn, temperate and tropical soybean, cotton, rice and sugarcane (Lawrence et al., 2018). Fertilization dynamics and annual fertilizer amounts in CLM5 depend on the crop functional types and vary spatially and yearly based on the land use and land cover change time series from the Land Use Model Intercomparison Project (Lawrence et al., 2019). Mineral fertilizer application starts during the leaf emergence phase of crop growth and continues for 20 days and manure nitrogen is applied at slower rates of 0.002 kg N m$^{-2}$ per year. For a more detailed description of the features and formulations of CLM5 the reader is referred to the technical description and latest literature (Lawrence et al., 2018, 2019a).

Line 255- the summary of weather patterns in 2017-2020 would be better captured in a figure

We provide more analysis of the weather patterns and differences between the different forcing products in the supplementary material, e.g.:

[Figure]

Figure S1: SEAS5 total monthly precipitation amounts from seasonal forecasts starting on the 1$^{st}$ of April (SEAS5-S) and sub-seasonal forecasts starting on the 1$^{st}$ of July (SEAS5-SUB) for the years 2017 and 2018,

for (a,b) the AUS-VIC domain and (c,d) the DE-NRW domain, compared to WFDE5 data for the respective domains.

**Results—Section 3.1** The title of the paper and much of the introduction focuses on Crop Yields. However, the first 2 pages of the results focus on soil moisture estimations. This is confusing to the reader. If these results are important (and it seems they are) both the title and intro material should also emphasize the soil moisture estimations as much as the crop yields.

We agree and have changed the title to:

Seasonal soil moisture and crop yield prediction with SEAS5 long-range meteorological forecasts in a land surface modelling approach

We have also restructured and rewritten several parts of the introduction accordingly.

(Line 127-131): In CLM5, crop productivity is a dynamic nonlinear interaction between meteorological conditions, crop phenology, nutrient dynamics, and water availability in the soil. Thus, a reliable prediction of the soil moisture regime is also essential for the relevance of land surface model applications for climate change research and is a major source of uncertainty for the simulation of the terrestrial carbon cycle (Trugman et al., 2018).